# OCTA measurements in Behcet's disease across different stages of the disease activity: A systematic review and meta-analysis

Mehrdad Mozafar[1,2‡]*, Mobina Amanollahi[1,3‡], Reza Samiee[1,3], Melika Jameie[4,5], Ali Mousavi[6], Zahra Ghanbari[1], Helia Nafar[1], Negar Mozafar[7], Fatemeh Amiri[1], Mehdi Azizmohammad Looha[8], Elias Khalili Pour[3]*, Nazanin Ebrahimiadib[9]

1 School of Medicine, Tehran University of Medical Sciences, Tehran, Iran, 2 Division of Vascular and Endovascular Surgery, Department of Surgery, Shohada-Tajrish Medical Center, Shahid Beheshti University of Medical Sciences, Tehran, Iran, 3 Translational Ophthalmology Research Center, Farabi Eye Hospital, Tehran University of Medical Sciences, Tehran, Iran, 4 Neuroscience Research Center, Iran University of Medical Sciences, Tehran, Iran, 5 Iranian Center of Neurological Research, Neuroscience Institute, Tehran University of Medical Sciences, Tehran, Iran, 6 School of Medicine, Tabriz University of Medical Sciences, Tabriz, Iran, 7 School of Medicine, Shahid Beheshti University of Medical Sciences, Tehran, Iran, 8 Basic and Molecular Epidemiology of Gastrointestinal Disorders Research Center, Research Institute for Gastroenterology and Liver Diseases, Shahid Beheshti University of Medical Sciences, Tehran, Iran, 9 Department of Ophthalmology, University of Florida, College of Medicine, Gainesville, Florida, United States of America

‡ Mehrdad Mozafar and Mobina Amanollahi carry a co-first author status.
* ekhalilipour@gmail.com (EK); mehrdad.mozafar98@gmail.com (MM)

## Abstract

### Purpose

Behcet disease (BD) is an autoimmune disease characterized by diffuse all-sized obliterative vasculitis, thrombotic vasculopathy, and common ocular involvement. This study aims to evaluate the retinal microvascular alterations in BD using optical coherence tomography angiography (OCTA).

### Methods

PubMed, Embase, and Web of Sciences were systematically searched for relevant studies assessing OCTA measurements in BD and healthy controls (HCs). Meta-analysis was conducted on OCTA parameters with at least two studies using the same OCTA device, with similar case and control groups. A qualitative synthesis approach was used to report the data that could not be pooled.

### Results

Twenty-eight related studies (769 BD subjects, 123 active BU eyes, 462 inactive BU eyes, 112 non-specified ocular BD, and 486 non-ocular BD eyes) were included. Patients with inactive Behcet uveitis (BU) and non-ocular BD showed a statistically significant greater FAZ size and lower superficial retinal capillary plexus and deep

**Data availability statement:** The data supporting the findings is available in a compressed Supporting information as Supporting information 2.

**Funding:** The author(s) received no specific funding for this work.

**Competing interests:** The authors have declared that no competing interests exist.

retinal capillary plexus vessel density (VD) in comparison to HCs, particularly in the parafoveal sector. Also, radial peripapillary capillary (RPC) VD was lower in patients with inactive BU than HCs. However, no significant difference in OCTA parameters was found in patients with active BU compared to HCs.

## Conclusions

Meta-analysis demonstrated reduced VD in various regions and greater FAZ size in superficial and deep retinal plexuses, particularly in inactive BU and non-ocular BD, which are mostly studied so far. However, more studies with larger sample sizes should draw a more definite conclusion. OCTA can provide valuable insights into retinal microvasculature changes in BD.

---

### 1. Introduction

Behcet's disease (BD) is characterized as a chronic, inflammatory condition that affects several organs such as skin, mucous membranes, eyes, joints, and lungs, as well as gastrointestinal, genital, and central nervous systems [1–4]. It is distinguished by immune-mediated vasculitis, which can affect blood vessels of various organs [5]. Ocular inflammation most commonly manifests as relapsing-remitting uveitis, affecting more than 70% of patients [6–9].

According to the European Alliance of Associations for Rheumatology (EULAR), multimodal imaging is recommended for managing patients with BD-associated severe eye disease. This is defined by the presence of retinal vasculitis affecting the macula or optic nerve leading to significant vision loss. Multimodal imaging typically includes fluorescein angiography (FA), indocyanine green angiography (ICGA), and spectral domain optical coherence tomography (SD-OCT) [10,11]. FA remains the gold standard for documenting and monitoring the involvement of the posterior segment of the eye in BD. However, it's an invasive procedure requiring an injected dye and has limitations in visualizing deeper vascular structures of the retina [6,12].

Optical coherence tomography angiography (OCTA) offers a new approach to visualize the intricate network of microvasculature in the retina and choroid. Unlike traditional angiography, OCTA does not require intravenous contrast injection [13,14]. OCTA can create detailed, layered images of the retinal vasculature, such as the superficial retinal capillary plexus (SRCP), the deep retinal capillary plexus (DRCP), and the choriocapillaris. The detection of retinal or choroidal vascular changes, measurement of the foveal avascular zone (FAZ), and quantification of vascular density (VD) in the inner retina, outer retinal circulation, or Choriocapillaris are all advantages of OCTA [14]. OCTA has been widely used to assess the microvasculature structure in various diseases such as diabetic retinopathy [15, 16], age-related macular degeneration [17], optic neuropathies [18], and autoimmune diseases such as systemic lupus erythematous [19], Neuromyelitis optica spectrum disorders (NMOSD) and Myelin oligodendrocyte glycoprotein antibody disease (MOGAD) [20].

Previously, the results of ocular and non-ocular BD retinal microvasculature were assessed by Ji et al. by pooling relevant studies up to 2021 [21]. However, several gaps still exist that need to be addressed. A comprehensive systematic review incorporating all the OCTA parameters including choriocapillaris and radial peripapillary capillary VD was missing. Furthermore, despite the typical categorization of ocular BD into those with active or inactive uveitis by various studies, the previous study pooled both of the results together, which could justify the higher amount of heterogeneity throughout their analyses. In fact, uveitis can lead to retinal vasculitis and hence result in microvasculature alterations in OCTA imaging [22]. Thus, it is imperative to separately evaluate ocular BD based on uveitis activity status, which is the primary goal of the present study. Herein, we aim to comprehensively review the studies that evaluated the microvascular changes in the eyes of patients with BD (i.e., active or inactive ocular BD and non-ocular BD) using OCTA.

## 2. Methods

This systematic review and meta-analysis adhere to the Preferred Reporting Items for Systematic Reviews and Meta-Analyses (PRISMA) guideline [23] (Tables S5 and S6 in S1 File). The study protocol was registered on the International Prospective Register of Systematic Reviews (PROSPERO) website (Registration code: CRD42023468476).

### 2.1. Search strategy

We searched PubMed, EMBASE, and Web of Sciences for relevant papers from the earliest to October 09, 2024. The keywords contain combinations of OCTA and Behcet headings as follows: ("optical coherence tomography angiography" OR "OCT angiography" OR "OCTA") AND ("Behcet"). The detailed search strategy for the three databases is provided in **Tables S1-3** in S1 File). Two independent reviewers did the initial screening (RS, MA), and disagreements were solved by consulting the senior author (EK). Included studies were not restricted to location, age, and gender discrepancies. We further explored the references of included studies for additional relevant articles.

### 2.2. Inclusion and exclusion criteria

Studies in which OCTA evaluated retinal microvasculature of patients with BD were incorporated into this review if they met the following criteria: (1) original peer-reviewed papers, (2) English language, (3) Including patients with BD, (4) Including a control group. Exclusion criteria were as follows: (1) reviews, conference abstracts, and case reports, (2) non-English, (3) non-human, (4) no comparative control group, and (5) not stating the presence of ocular involvement in BD patients. The detailed selection process is illustrated in **Fig 1**.

### 2.3. Data extraction

Following demographics and clinical information were collected from each article: author's name, publication year, study design, status of ocular inflammation (active Behcet's uveitis (BU), inactive BU, non-ocular Behcet), sample size, female percentage, age, disease duration, studies' inclusion and exclusion criteria, axial length, intraocular pressure (IOP) of enrolled eyes, and best corrected visual acuity (BCVA). We also extracted OCTA-related data such as device type, software, field of view, VD in the area of SRCP, DRCP and their relative subregions (whole, foveal, parafoveal, perifoveal), Choriocapillaris flow area, radial peripapillary capillary (RPC). Two independent authors extracted the data (MM, MA), and any conflicts were resolved by the senior advisor.

Active BU was defined by included studies as clinical signs of active posterior or panuveitis on ophthalmologic examination [24–29], whereas inactive BUs were selected from patients who had a history of previous uveitis attacks without apparent anterior or posterior uveitis at the time of study [30–33]. In the context of non-ocular BD, patients were identified as those without signs of ocular involvement. This includes an absence of past inflammatory signs such as posterior synechiae, vitreous cells, vascular sheathing, and chorioretinal scars [28, 34–40]. Additionally, some studies did not specify the activity status of ocular involvement [37, 40]. We referred to this population as a non-specified ocular BD group.

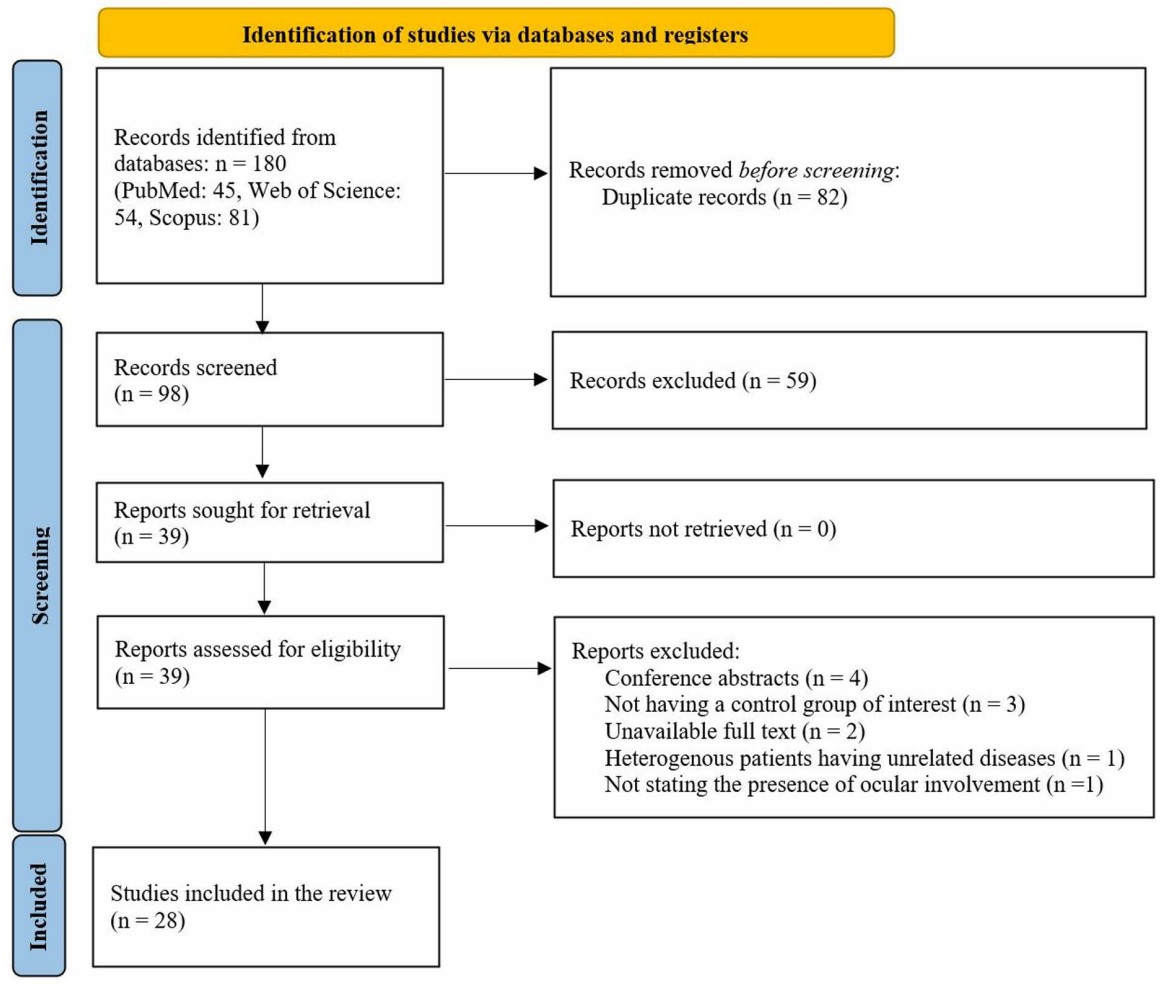

**Fig 1. PRISMA flowchart describing the process of search, screening, inclusion, and exclusion.**

### 2.4. OCTA parameters

The primary metric analyzed in these studies was VD. It represents the proportion of a retinal area occupied by blood vessels. Researchers measured VD in various retinal layers and regions throughout the studies. However, the wording for superficial and deep layers varied slightly between studies. To ensure consistency, we categorized these layers simply as either the SRCP or the DRCP. Importantly, we confirmed that the definitions of all reported layers and regions aligned with established standards. Furthermore, to minimize potential discrepancies caused by differences in imaging devices, we only compared VD measurements from studies using the same type of OCTA device. The Early Treatment Diabetic Retinopathy Study (ETDRS) grid, established in 1991, provides a standardized approach for segmenting the macula into specific regions like whole, foveal, parafoveal, and perifoveal. This grid is widely used when analyzing vessel density in the SRCP and DRCP layers of OCTA scans [41]. Thus, we referred to the foveal as the central macula circle (1 mm in diameter), parafoveal as the inner ring circle (3 mm in diameter), and perifoveal as the outer ring (6 mm in diameter) [41]. Some studies subdivided the regions into superior, inferior, nasal, and temporal. We also analyzed the whole image retinal layer's VDs in the 3*3 and 6*6 field of view separately, for a more precise comparison.

**SRCP** is defined as a retinal layer extended from 3 μm below the inner limiting membrane (ILM) to 15 μm below the inner plexiform layer (IPL) [25,27–30, 32, 34, 37, 42–46]. SRCP was measured in 3*3 fields of view by six studies [25,27,33,37,42,44], and in 6*6 fields of view by thirteen studies [24,26,29,34–36,38,43,45,47–50].

**DRCP** is generally defined as the retinal layer spanning from IPL to the outer plexiform layer (OPL), with minor variations in the exact boundaries across the studies included in this review [25,27–30,32,34,37,42–46]. Six studies measured DRCP in 3*3 fields of view [25, 27, 33, 37, 42, 44], while in twelve studies, DRCP was viewed in 6*6 fields of view [24, 26, 29, 34–36, 43, 45, 47–50].

**Choriocapillaris** is a thin layer beneath the Bruch membrane [51]. Unlike other layers, the flow area is its general measurement metric, which can be measured in a 1 mm or 3 mm radius by twelve studies [24, 29–31, 34, 35, 43, 45, 47–50]. To reduce the risk of heterogeneity, we analyzed the choriocapillaris in 1 mm and 3 mm radius separately.

**FAZ** is a capillary-free area centered on the fovea. Its size is measured in square millimeters (mm²) and reflects the circulation status in various diseases [52]. It makes the most common variable reported by 26 included studies [25, 27–40, 42–50, 53]. Reportedly, previous studies demonstrated greater FAZ parameters in the superficial layer than in the deep layer; highlighting separate measurements of FAZ based on the retinal layer [54]. The included studies in the current review mostly reported the FAZ in the superficial layer [25, 27, 30–35, 38, 39, 45, 53], while some reported the metric in the deep layer [25, 27, 30, 32–34, 39, 43, 53]. Noteworthy, some did not specify the layer in which FAZ was measured [28, 29, 35, 37, 40, 42, 44, 45, 47–50]; hence we referred to those as FAZ in general and were analyzed separately.

**RPC** is the primary OCTA-related measurement of the optic disc, the vascular plexus in the retinal nerve fiber layer (RNFL). In the present review, seven studies measured RPC VD and defined it as the area that prolongs from the ILM to the RNFL [29, 31, 45, 47–50].

## 2.5. Quality assessment

Newcastle-Ottawa Scale (NOS) was used as a risk-of-bias assessment tool [55]. Studies with less than five scores are considered at high risk of bias, and those with more than five are considered "good" ones. Two authors (MM, MA) reviewed the included studies for this purpose, and disagreements were referred to the senior author (EK).

## 2.6. Statistical analysis

This meta-analysis employed a quantitative approach (synthesis) to analyze all measurable OCTA imaging outcomes. The analysis generated mean and standard deviation (SD) values for each outcome. It's important to note that a qualitative approach (synthesis) was used to report findings for parameters that could not be combined in specific comparisons (**Supplemental Results** in S1 File)). Whenever possible, values reported using other metrics were converted to mean and SD for consistency.

Quantitative data synthesis was conducted on all measurements assessed in at least two studies. These studies needed to share similar patient characteristics (active or inactive BU and non-ocular BD) and utilize the same type of OCTA device. Exceptions were made when a qualitative approach was deemed necessary to report study findings. The False Discovery Rate (FDR) method was applied to correct for multiple comparisons. Stata version 16 software was used for the statistical analyses. To compare cases and controls for each OCTA outcome, the analysis employed Hedges' g, a standardized mean difference statistic, along with a 95% confidence interval (CI) to represent the effect size. Heterogeneity, or variation, across studies, was assessed using Higgin's $I^2$ statistic. A fixed-effects model was used when heterogeneity was below 40%, while a random-effects model was used for studies with higher heterogeneity. The data supporting the findings is provided in S2 File).

## 3. Results

### 3.1. Studies' characteristics

Twenty-eight studies were included. **Table 1** illustrates an overview of the studies' characteristics. 769 BD subjects (112 non-specified eyes, 123 active BU eyes, 462 inactive BU eyes, and 486 non-ocular BD eyes) and 1100 healthy control

**Table 1. Overview of demographic and baseline variables among included studies.**

| Study (year) | Behcet criteria | Trait | Subjects (eyes) | Female (n) | Age (years) (mean±SD) | Disease duration (years) (mean±SD) | AL (mm) (mean±SD) | BCVA (logMAR) (mean±SD) | IOP (mean±SD) | Type of OCTA |
|---|---|---|---|---|---|---|---|---|---|---|
| Accorinti et al. (2019) [24] | ISGBD | Active BU / Inactive BU / HC | 8 (8) / 15 (15) / 15 (15) | 3 / 5 / 8 | 40±15.2 / 38.7±13.2 / 35.7±7.8 | 10.1±9.3 / 9±9 / - | - | 0.2±0.3 / 0±0.1 / 0 | - | PLEX Elite |
| Aksoy et al. (2020) [30] | ISGBD | Inactive BU / HC | 35 (35) / 30 (30) | 17 / 15 | 38±7.1 / 37±8.2 | 2.5±1.9 / - | - | - | - | RTVue XR Avanti |
| Balicoglu Yilmaz et al. (2020) [31] | ISGBD | Inactive BU / HC | 20 (40) / 26 (52) | 8 / 16 | 42.7±12.02 / 44.46±14.46 | - | - | 0.1 (0–3) (R) / 0.18 (0–3) (L) (median (min-max)) / - | - | RTVue XR Avanti |
| Cheng et al. (2018) [25] | ISGBD | Active BU / HC | 19 (19) / 25 (25) | 5 / 10 | 37.5±6.5 / 38.7±9.5 | 2.7±2 / - | - | 0.19±0.24 / 0 | - | RTVue XR Avanti |
| Comez (2019) [34] | ISGBD | Non-ocular BD / HC | 21 (42) / 20 (40) | 10 / 10 | 39.81±8.93 / 41.21±9.87 | 6.04±5.07 / - | - | 0 / 0 | 14.75±3.46 / 15.19±3.93 | RTVue XR Avanti |
| Dai et al. (2024) [56] | ISGBD | Inactive BU / HC | 52 (52) / 50 (50) | 13 / 21 | 31.97±10.03 / 37.44±9.17 | 3.79±2.79 / - | - | 0.42±0.49 / 0.03±0.05 | 15.10±2.73 / 15.62±2.57 | Svision |
| Degirmenci et al. (2018) [53] | ISGBD | Non-Ocular BD / HC | 23 (44) / 29 (49) | 11 / 13 | 45.7±5.1 / 51.4±7.1 | - | - | -0.1±0.02 / -0.12±0.01 | 18.1±2.3 / 19.3±1.9 | RTVue XR Avanti |
| Emre et al. (2019) [26] | ISGBD | Active BU / HC | 16 (26) / 15 (30) | 10 / 10 | 39.44±13.56 / 38.1±6.76 | 14±11.7 / - | - | 0.2±00.1 / 0 | 14.6±2.8 / 13.9±2.5 | RTVue XR Avanti |
| Eser-Ozturk et al. (2021) [32] | ISGBD | Inactive BU / HC | 22 (42) / 19 (38) | 4 / 8 | 35.73±11.76 / 40.1±9.08 | 7.36±6.12 / - | - | 0.24±0.32 / - | - | DRI OCT Triton plus |
| Ferreira et al. (2023) [42] | ISGBD | Inactive BU / Non-Ocular BD / HC | 14 (23) / 13 (26) / 13 (26) | 7 / 6 / 7 | 40.6±11.7 / 41.2±8.7 / 39.9±11.1 | 12.2±6.2 / 15.6±8.7 / - | - | 0 (0–0.7) (median, range) / 0 (0–0) / 0 (0–0) | - | Heidelberg |
| Goker et al. (2019) [35] | ISGBD | Non-ocular BD / HC | 11 (22) / 14 (28) | 7 / 8 | 37.5±14.3 / 39.2±14.9 | 10.4±6.62 / - | 22.85±0.26 / 22.96±0.35 | - | 16.14±2.36 / 16.18±2.24 | RTVue XR Avanti |
| Guo et al. (2023) [43] | ISGBD | Inactive BU / HC | 33 (57) / 35 (60) | 10 / 13 | 34±8.87 / 31±10.23 | 5.42±3.45 / - | - | - | - | BM-400K BMizar |
| Kianersi et al. (2024) [49] | - | Non-specified Ocular BD / Non-ocular BD / HC | - (49) / - (18) / - (43) | 14 / - / 15 | 42.23±10.7 / - / 46.16±12.8 | - | 22.9±0.7 / - / 23.03±0.6 | 0.74±0.25 / - / 0.80±0.18 | 15.5±2.5 / - / - | RTVue XR Avanti |
| Khairallah et al. (2017) [27] | ISGBD | Active BU / HC | 25 (44) / 11 (22) | 3 / - | 31.8±9 / - | - | - | 0.62±0.57 / - | - | DRI OCT Triton plus |
| Koca et al. (2019) [37] | ISGBD | Non-specified Ocular BD / Non-ocular BD / HC | - (43) / - (51) / 53 (53) | - / - / - | 40.95±8.9 / 40.2±9.9 / 41.59±8.9 | 8.1 [3.8–11] / 11.6 [5–16] / - | - | 0 (0–0.7) (median) / 0.97±0.09 (decimal) / - | - | RTVue XR Avanti |
| Karalezli et al. (2021) [47] | ISGBD | Non-Ocular BD / HC | 28 (56) / 25 (50) | 18 / 15 | 38.5±14.3 / 40.2±14.1 | 7.4±3,6 / - | 22.51±2.2 / 23.01±1.2 | - | 14.12±2.2 / 14.22±2.4 | RTVue XR Avanti |
| Küçük et al. (2022) [48] | ISGBD | Non-Ocular BDt / HC | 33 (56) / 33 (61) | 17 / 18 | 40.03±7.49 / 38.79±9.93 | 7.04±3.97 / - | 23.47±0.83 / 23.19±0.8 | 0.004±1.52 / 0 | 16.17±2.38 / 15.44±2.33 | RTVue XR Avanti |

*(Continued)*

**Table 1.** (Continued)

| Study (year) | Behcet criteria | Trait | Subjects (eyes) | Female (n) | Age (years) (mean±SD) | Disease duration (years) (mean±SD) | AL (mm) (mean±SD) | BCVA (logMAR) (mean±SD) | IOP (mean±SD) | Type of OCTA |
|---|---|---|---|---|---|---|---|---|---|---|
| Karaca et al. (2023) [36] | – | Inactive BU<br>Non-ocular BD<br>HC | 22 (37)<br>26 (48)<br>22 (44) | 10<br>11<br>10 | 41.9±9.4<br>41±11.6<br>44±14.2 | 11.7±7.8<br>10.6±8.3<br>– | – | – | – | RTVue XR Avanti |
| Nassar et al. (2022) [46] | | Inactive BU<br>HC | 10 (10)<br>20 (20) | – | – | – | – | 0.4<br>1 | – | RTVue XR Avanti |
| Pei et al. (2019) [44] | ISGBD | Inactive BU<br>HC | 60 (102)<br>62 (124) | 15<br>20 | 29±7.4<br>29±10.18 | – | – | – | – | RTVue XR Avanti |
| Raafat et al. (2019) [38] | ISGBD | Non-ocular BD<br>HC | 10 (20)<br>10 (20) | 3<br>– | 36.6±9.54 | 12.7±8.8<br>– | – | 0.01±0.09<br>– | 13.45±2.09<br>– | RTVue XR Avanti |
| Shen et al. (2024) [50] | | Active BU<br>HC | 25<br>416 (500) | –<br>246 | –<br>45.66±18.18 | – | – | – | – | RTVue XR Avanti |
| Smid et al. (2021) [28] | ISGBD | Active BU<br>Non-ocular BD<br>HC | 23 (21)<br>23 (23)<br>22 (22) | 11<br>11<br>8 | 51±10<br>48±14<br>44±13 | 17±9<br>11±8<br>– | – | – | – | Heidelberg |
| Simsek et al. (2022) [39] | ISGBD | Non-ocular BD<br>HC | 38 (38)<br>35 (35) | 12<br>14 | 34.6±5.7<br>33.5±5.1 | 7.1±2.5<br>– | 22.76±0.33<br>23.04±0.3 | 0<br>0 | 15.25±2.89<br>14.66±2.35 | RTVue XR Avanti |
| Türkcü et al. (2020) [33] | – | Inactive BU<br>HC | 30 (30)<br>31 (31) | 7<br>10 | 33.8±4.51<br>32.6±4.06 | 3.03±2.63<br>– | – | – | – | RTVue XR Avanti |
| Yan et al. (2021) [29] | ISGBD | Active BU<br>HC | 13 (24)<br>15 (24) | 6<br>10 | 32.9±15.2<br>38.5±12.4 | – | – | 0.87±0.62<br>– | – | RTVue XR Avanti |
| Yılmaz et al. (2021) [40] | ISGBD | Non-specified Ocular BD<br>Non-ocular BD<br>HC | 20 (20)<br>20 (20)<br>30 (30) | 11<br>16<br>17 | 36±12.2<br>40.1±10.7<br>39.6±9.6 | 10.9±7<br>9.7±7.3<br>– | – | 0.2±0.3<br>0.01±0.05<br>0±0 | – | RTVue XR Avanti |
| Yılmaz Tuğan et al. (2022) [45] | ISGBD | Non-ocular BD<br>HC | 22 (22)<br>24 (24) | 8<br>8 | 13.75±2.25<br>14.09±3.27 | 1.73±0.65<br>– | 23.27±0.58<br>23.42±0.59 | – | 15.63±1.14<br>16.63±1.49 | RTVue XR Avanti |

*Abbreviations*: HC, Healthy control; ISGBD, International study group for Behcet disease; AL, Axial length; BCVA, Best corrected visual acuity; IOP, Intraocular pressure; OCTA, Optical coherence tomography angiography; BU, Behcet uveitis; BD, Behcet disease.

(HC) subjects (1446 HC eyes) were analyzed for different parameters. The majority of the patients and HCs were male (58.7% across all studies) and in their 40s. One of the studies addressed patients' right and left eyes; hence, we considered this study as two distinct ones when conducting a meta-analysis (two spreadsheet rows were dedicated to this study) [31]. Among the studies, six assessed active BU eyes [24, 26–29, 50], 11 recruited inactive BU eyes (during the remission period) [24, 25, 30–33, 36, 42–44, 46, 56], and 14 included BD patients with no present or history of uveitis (non-ocular BD) [28,34–40,42,45,47–49,53]. Two did not determine the active or past BU and were allocated as non-specified ocular BD [37, 40, 49]. Additionally, the studies' inclusion and exclusion criteria for selecting cases and controls were demonstrated in **Table 2**.

Except for FAZ, all the analyses regarding different parameters (VD or flow area) were grouped based on the OCTA device because of the basic differences in the values. Twenty-one studies applied RTVue XR Avanti (Optovue Inc., Fremont, CA) along with AngioVue or AngioAnalytics software [25, 26, 29–31, 33–40, 44–50, 53], two used Topcon DRI OCT Triton Plus along with ImageJ or IMAGEnet software (Topcon Corp., Tokyo, Japan) [27, 32], and two used Heidelberg Spectralis OCTA (Heidelberg Engineering, Heidelberg, Germany) with internal Heidelberg software [28, 42]. Moreover, a study used PLEX Elite [24], one used BM-400K Bmizar [43], and one conducted OCTA with Svision [56] (**Table 3**).

### 3.2. Metrics

**Table 3** represents the retinal layers and related regions evaluated by studies. Twenty-six studies measured FAZ size [25, 27–40, 42–50, 53]. The included studies considered VD as their measurement (rather than perfusion density (PD) or fractal dimension (FD)) across OCTA variables. All of the studies measured macular VD (superficial and deep layers) [24–40, 42–50, 53,56], which was the most common site measured among the studies. Only seven assessed RPC VD [29,31,45,47–50]. The included studies assessed macular parafoveal VD and RPC VD as the most common variables (**Table 3**). A summary of the study results is illustrated in **Fig 2**. In the following, we demonstrate the results of meta-analyses conducted on the included studies (Quantitative data synthesis) (**Table 4**). Noteworthy, each study's detailed results are presented in **Table 5**. Moreover, the qualitative synthesis of the included studies is reported as **Supplemental Results** in **S1 File**).

### 3.3. Active BU vs. HC

**3.3.1. Whole (6\*6) SRCP VD** (active BU vs. HC)**.** Two studies compared SRCP whole (6\*6) VD between active BU and HC subgroups using the RTVue XR Avanti device [26, 29]. The pooled results (50 active BU and 54 HC eyes) showed no significant difference between the two groups (Hedges g = −0.59, CI= [−2.07 to 0.88], I$^2$ = 74.28%, P value = 0.43, Corrected P value = 0.43) (**Fig 3A**).

**3.3.2. Whole (6\*6) DRCP VD** (active BU vs. HC)**.** Two studies compared DRCP whole (6\*6) VD between active BU and HC subgroups using the RTVue XR Avanti device [26, 29]. The pooled results (50 active BU and 54 HC eyes) showed no significant difference between the two groups (Hedges g = −0.36, CI= [−1.12 to 0.40], I$^2$ = 74.28%, P value = 0.36, Corrected P value = 0.36) (**Fig 3B**).

**3.3.3. FAZ** (active BU vs. HC)**.** Two studies compared FAZ between active BU and HC subgroups using the RTVue XR Avanti [29] and Heidelberg [28] devices. Meta-analysis (45 active BU and 46 HC eyes) revealed no significant difference between the two groups (Hedges g = 0.853, CI= [−0.05 to 1.76], I$^2$ = 77.39%, P value = 0.06, Corrected P value = 0.06) (**Fig 3C**).

### 3.4. Inactive BU vs. HC

**3.4.1. Whole (3\*3) SRCP VD** (inactive BU vs. HC)**.** Four studies evaluated the whole picture (3\*3) SRCP VD in inactive BU and HC eyes using the RTVue XR Avanti device [25, 33, 46, 57]. (59 active BU and 76 HC eyes) revealed no significant

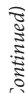

**Table 2. Exclusion and inclusion criteria of each study.**

| Study | Ophthalmological exclusion criteria | Health conditions exclusion criteria | Patient's Inclusion criteria | HC inclusion criteria |
|---|---|---|---|---|
| Accorinti et al. (2019) [24] | 1. Spherical refractive error ≥ ± 6 diopters<br>2. Astigmatism ≥ ± 3 diopters<br>3. Presence of significant media opacities (e.g., cataract or corneal opacity)<br>4. Retinal and optic nerve diseases<br>5. Any anterior segment surgery performed less than 6 months prior to the examination<br>6. Previous vitreoretinal surgery or trauma | 1. Age < 16 years old and age > 70 years old | 1. Diagnosis of ocular Behcet based on the ISGBD<br>2. At least one episode of posterior uveitis or panuveitis characterized by the presence of either retinal infiltrate, retinal vasculitis, optic nerve, and/or macular signs of inflammation | 1. No ocular disease<br>2. Spherical refractive error ≤ ± 6 diopters<br>3. Astigmatism ≤ ± 3 diopters<br>4. Absence of significant media opacities |
| Aksoy et al. (2020) [30] | 1. Ocular Behcet with anterior uveitis only<br>2. Spherical equivalent refractive error greater than ± 2.0 diopters 3. Substantial media opacities<br>4. Trauma<br>5. Retinal diseases such as diabetic retinopathy, hypertensive retinopathy, central serous chorioretinopathy, and macular degeneration<br>6. Optic nerve diseases such as glaucoma or optic neuropathy | – | 1. In the remission phase of BD and fulfilling ISGBD criteria<br>2. Spherical refractive error within ± 2.0 diopters<br>3. Cylindrical error within ± 2.0 diopters | 1. No ocular or systemic diseases<br>2. Spherical refractive error within ± 2.0 diopters and a cylindrical error within ± 2.0 diopters |
| Balicoglu Yilmaz et al. (2020) [31] | 1. Significant media opacities such as dense cataracts, vitreous haze, or vitreous hemorrhage<br>2. Poor fixation that did not allow obtaining images of adequate quality<br>3. Intraocular pressure > 21 mmHg<br>4. Refractive error greater than 3 diopters<br>5. Confounding ocular pathology such as diabetic retinopathy, hypertension, glaucoma, or optic neuropathy | – | 1. Diagnosis of BU based on ISGBD<br>2. stable clinical remission | – |
| Cheng et al. (2018) [25] | 1. Anterior uveitis only<br>2. Spherical equivalent refractive error greater than 66.0 diopters<br>3. Significant cataract or vitreous opacity, diabetic retinopathy, age-related macular degeneration, and glaucoma<br>4. History of intraocular surgery within the previous year | – | 1. BD in remission<br>2. Fulfilling the diagnosis criteria of ISGBD | 1. No history of systemic disease, excluding ocular disease (e.g., diabetes retinopathy or glaucoma)<br>2. BCVA < 20/20<br>3. Spherical equivalent refractive error of more than 66.0 diopters |
| Comez et al. (2019) [34] | 1. Active or previous findings of uveitis (ciliary injection, anterior chamber inflammatory cells, keratic precipitations, papillitis, inflammatory cells in the vitreous or retinal vasculitis)<br>2. Additional ocular or systemic disease (glaucoma, diabetic retinopathy, etc.)<br>3. History of topical or systemic medication<br>4. Ocular surgery<br>5. Treatment for uveitis<br>6. Good-quality images could not be obtained with OCTA | – | 1. Diagnosis of non-ocular BD according to ISGBD<br>2. No active or previous uveitis findings | – |
| Dai et al. (2024) [56] | 1. Ocular surgery and other ocular diseases<br>2. Refractive media opacity (e.g., corneal opacity and cataract)<br>3. Retinal disorders (e.g., macular degeneration and central serous chorioretinopathy)<br>4. Optic nerve diseases (e.g., glaucoma and optic neuropathy) | 1. Systemic diseases that possibly affect the OCTA results such as systemic diseases (e.g., diabetes mellitus, hypertension, and Alzheimer's disease) | 1. Fulfilling the ISGBD criteria<br>2. Absence of intraocular inflammation (e.g., anterior chamber inflammation, vitreous haze, and active retinal lesions) for at least three months | 1. Absence of any abnormal medical history |

*(Continued)*

| Study | Ophthalmological exclusion criteria | Health conditions exclusion criteria | Patient's Inclusion criteria | HC inclusion criteria |
|---|---|---|---|---|
| Degirmenci et al. (2018) [53] | 1. Other forms of ocular pathology | 1. Systemic hypertension, and diabetes mellitus | 1. Fulfilling the ISGBD criteria | – |
| Emre et al. (2019) [26] | 1. Significant media opacities<br>2. Trauma<br>3. Other retinal diseases such as retinal vascular occlusive disease unrelated to BD, diabetic retinopathy, hypertensive retinopathy, central serous chorioretinopathy, and macular degeneration<br>4. Optic nerve diseases such as glaucoma or optic neuropathy | – | 1. Fulfilling the ISGBD criteria | – |
| Eser-Ozturk et al. (2021) [32] | 1. Other retinal and optical nerve diseases, such as diabetic retinopathy, hypertensive retinopathy, macular degeneration, glaucoma, and optic neuropathy<br>2. Cystoid changes in the FAZ area | – | 1. BU in clinical remission diagnosed according to ISGBD | – |
| Ferreira et al. (2023) [42] | 1. Isolated anterior uveitis<br>2. Spherical equivalent refractive error greater than 6.0 diopters<br>3. Media opacity<br>4. Retinal or optic disk diseases not related to BD such as diabetic retinopathy, macular degeneration, and glaucoma<br>5. Intraocular surgery <6 months<br>6. Intake of medication that could affect the macula | 1. Systemic diseases | 1. Age older than 18 years<br>2. Inactive BU diagnosed by ISGBD | – |
| Goker et al. (2019) [35] | 1. Substantial media opacity<br>2. AL >24 mm and <22<br>3. IOP readings greater than 21 mmHg<br>4. Previous intraocular or periocular interventions, and intraocular operations within 12 months before study enrollment<br>5. Histories of acute or chronic red eye and symptoms of inflammation including cells in the anterior chamber and/or vitreous cavity, posterior synechia, keratic precipitates on the corneal endothelium, iris pigments on the anterior lens capsule, vascular sheathing, or optic disc pallor<br>6. Subclinical optic disc staining and retinal capillary leakage in early and late-phase FA images<br>7. Retina pigment epithelium mottling, macular edema in OCT, and specific visual field defects in visual field test | 1. Other systemic abnormalities (e.g., hypertension or diabetes mellitus) | 1. Diagnosis based on ISGBD<br>2. Normal results in the biomicroscopic evaluation<br>3. BCVA of at least 20/20 | 1. No systemic or ocular diseases<br>2. Refractive spherical or cylindrical error <2 diopters<br>3. Visual acuity of 20/20 in both eyes |
| Guo et al. (2023) [43] | 1. Obvious media opacities,<br>2. Trauma<br>3. Refractive error higher than 4 diopters<br>4. History of retinal laser photocoagulation<br>5. Other retinal diseases | – | 1. Accordance with ISGBD criteria<br>2. Inactive BU defined as the absence of intraocular inflammation for at least 3 months | – |
| Kianersi et al. (2024) [49] | 1. History of eye surgery<br>2. Concurrent retinal conditions<br>3. Eye injuries<br>4. Other ocular that could affect the macula | 1. Systemic diseases that could affect the macula. | – | 1. Patients referred for glasses fitting with a refractive error of less than 2 diopters |

*(Continued)*

**Table 2.** (Continued)

| Study | Ophthalmological exclusion criteria | Health conditions exclusion criteria | Patient's Inclusion criteria | HC inclusion criteria |
|---|---|---|---|---|
| Khairallah et al. (2017) [27] | 1. Significant media opacities<br>2. Trauma<br>3. Other retinal diseases such as retinal vascular occlusive disease unrelated to BD, diabetic retinopathy, hypertensive retinopathy, central serous chorioretinopathy, and macular degeneration<br>4. Optic nerve diseases such as glaucoma or optic neuropathy | — | 1. Diagnosis based on ISGBD<br>2. Active BU (presence of vitritis in association with retinal vascular sheathing, retinal vascular leakage on FA, retinal infiltrates, optic disc edema, or anterior uveitis) | — |
| Koca et al. (2019) [37] | 1. Under 18 years old<br>2. History of ocular trauma<br>3. Significant media opacities<br>4. Retinal microvascular diseases (such as diabetic retinopathy)<br>5. Prior history of laser<br>6. Optic nerve and macular diseases such as glaucoma or optic neuropathy<br>7. Poor fixation for fundus evaluation | 1. Other systemic diseases | 1. Diagnosis based on ISGBD | — |
| Karalezli et al. (2021) [47] | 1. Previous surgery<br>2. Signs of recent or previous anterior or posterior uveitis<br>3. History of red eye<br>4. Pathology detected in biomicroscopic and fundus examination<br>5. Eyes with AL greater than 24 mm and shorter than 22 mm<br>6. IOP higher than 21 mmHg<br>7. History of topical or systemic medication<br>8. Optic disc staining and early, late-stage peripheral leakage on fundus FA<br>9. Poor visual quality in OCTA | 1. Systemic disease (hypertension, diabetes mellitus) | 1. BD with no ocular involvement | 1. Refractive index of less than 2 diopters<br>2. No systemic or ocular disease |
| Küçük et al. (2022) [48] | 1. Retinal changes not related to BD (e.g., age-related macular degeneration, chorioretinal scars, and mottling of the retinal pigment epithelium)<br>2. Optic nerve pathologies (e.g., glaucomatous optic nerve changes and optic nerve edema or pallor)<br>3. Ocular pathologies that reduce OCTA image quality (e.g., ocular surface disease and cataracts)<br>4. Glaucoma suspect based on a family or personal history<br>5. Subclinical optic disc staining and retinal capillary leakage in early, and late-phase FFA undertaken with the suspicion of retinal pathologies that may be associated with BD | 1. Systemic comorbidity other than BD affecting vascular (e.g., diabetes mellitus and hypertension) and neurological (e.g., multiple sclerosis and Alzheimer's disease) structures<br>2. Smoker | 1. Diagnosis based on ISGBD<br>2. OCTA image quality of 0.6 and above<br>3. No media opacity that could impair image quality<br>4. AL greater than 22.5 mm and less than 25 mm<br>5. Refractive error of ± 5 diopters spheres and ± 3 diopters cylinders | 1. Aged over 18<br>2. Did not have any eye or systemic disease<br>3. Absent regular use of systemic and ocular drugs<br>4. Without a history of eye surgery<br>5. Non-smokers<br>6. Did not have a family history of glaucoma<br>7. Normal optic disc appearance<br>8. OCTA image quality of 0.6 and above<br>9. No media opacity that could impair image quality<br>10. AL greater than 22.5 mm and less than 25 mm<br>11. Refractive error of ± 5 diopters spheres and ± 3 diopters cylinders |

*(Continued)*

| Study | Ophthalmological exclusion criteria | Health conditions exclusion criteria | Patient's Inclusion criteria | HC inclusion criteria |
|---|---|---|---|---|
| Karaca et al. (2023) [36] | 1. History of ocular trauma, diseases that may affect macula or optic disc<br>2. Poor fixation<br>3. Significant media opacity that may affect the quality of OCTA scans | 1. Systemic disease other than BD | 1. The first patient's group consists of Inactive BU without macular edema<br>2. The second patient group consists of BD subjects without ocular involvement with normal biomicroscopic and FA examination<br>3. Refraction errors of the patients were between −1.50 and +1.50 diopters | 1. No systemic or ocular disease |
| Nassar et al. (2022) [46] | 1. Dense media opacity<br>2. Trauma<br>3. Other retinal diseases causing uveitis<br>4. Retinal vascular occlusive disease unrelated to BD<br>5. Diabetic retinopathy<br>6. Optic nerve diseases such as glaucoma or optic neuropathy | 1. Hypertension | 1. Diagnosis based on ISGBD | – |
| Pei et al. (2019) [44] | 1. Scan quality index value less than 6<br>2. Significant media opacities<br>3. Trauma<br>4. Glaucoma<br>5. Other retinal diseases such as retinal vascular occlusive unrelated to BD, diabetic retinopathy, hypertensive retinopathy, central serous chorioretinopathy unrelated to BD, macular degeneration | – | 1. BU fulfilling ISGBD | 1. No previous ophthalmologic history |
| Raafat et al. (2019) [38] | 1. History of acute red eye and signs of recent or previous anterior or posterior uveitis (including any ocular signs of inflammation, including cells or flare, any point of synechiae, lens opacities or pigment in the anterior lens capsule, vitreous cells, signs of previous vasculitis, vascular sheathing or attenuation, chorioretinal scars or mottling of the retinal pigment epithelium, optic nerve edema, or pallor)<br>2. Previous intraocular or periocular interventions<br>3. Past or current history of topical treatments for glaucoma | 1. Systemic disease<br>2. Intake of medication that could affect the macula | 1. Non-ocular BD fulfilling ISGBD criteria | – |
| Shen et al. (2024) [50] | 1. Infectious uveitis<br>2. Diabetic retinopathy<br>3. Retinal vascular occlusion<br>4. Age-related macular degeneration<br>5. masquerade syndrome<br>6. Glaucoma<br>7. High myopia<br>8. Inability to maintain fixation<br>9. Significant media opacities | 1. Obesity<br>2. Hyperlipidemia<br>3. Heart disease,<br>4. Diabetes<br>5. Hypothyroidism<br>6. Chronic kidney disease<br>7. Metabolic syndrome<br>8. Liver diseases<br>9. Extreme low-carbohydrate or high-fat diets<br>10. Unhealthy lifestyle habits, including smoking, excessive drinking and chronic stress | – | 1. Devoid of any history of ocular inflammation, ocular injury, ocular surgery, or notable ocular pathologies |

*(Continued)*

**Table 2.** (Continued)

| Study | Ophthalmological exclusion criteria | Health conditions exclusion criteria | Patient's Inclusion criteria | HC inclusion criteria |
|---|---|---|---|---|
| Smid et al. (2021) [28] | 1. Refractive error higher than 4 diopters | — | 1. BU fulfilling ISGBD | — |
| Simsek et al. (2022) [39] | 1. Uveitis attack, including inflammatory cells in the anterior chamber, keratic precipitates, posterior synechia, iris pigments on the anterior lens capsule, vitreous opacities, optic disc pallor or optic atrophy, retinal vascular sheathing or sclerosis, and neovascularization of the disc or elsewhere<br>2. Ocular/orbital trauma<br>3. Any previous intraocular operation or laser intervention<br>4. A prior history of any type of optic neuropathies<br>5. Any type of retinal or choroidal diseases<br>6. Spherical and/or cylindrical refractive error> 3 diopters<br>7. AL>24 mm and < 22 mm<br>8. BCVA<20/20 | 1. Any systemic hematological or immunological disorders<br>2. Using medication<br>3. Smoking<br>4. Alcohol or drug abuse that might have induced vascular dysfunction | 1. Non-ocular BD fulfilling the ISGBD criteria | 1. No other ocular or systemic disease conditions |
| Türkcü et al. (2020) [33] | 1. Glaucoma, cataract, or diabetic retinopathy, macular abnormalities such as foveal atrophy which could affect the results dramatically or slightly diffuse macular edema | — | 1. Inactive BU | — |
| Yan et al. (2021) [29] | 1. Retinal and or optic nerve diseases other than OB<br>2. Intraocular surgery other than for uncomplicated cataract | — | 1. active BU fulfilling ISGBD criteria<br>2. Presenting with posterior uveitis or panuveitis characterized by retinal vasculitis and manifestations of optic nerve and/or macular inflammation | — |
| Yilmaz et al. (2021) [40] | 1. Other retinal and or optic nerve diseases (diabetic retinopathy, hypertensive retinopathy, central serous chorioretinopathy, macular degeneration, glaucoma, optic neuropathy, etc.)<br>2. Cystoid macular edema<br>3. Refractive errors> ±3.0 diopters of spherical equivalence<br>4. History of ocular trauma or surgery within the previous 3 months | 1. Systemic comorbidity that could affect the retinal microvasculature, such as diabetes mellitus and hypertension | 1. Non-ocular Behcet and active BU patients fulfilling the ISGBD criteria | 1. Refractive error <3.0 diopters of spherical equivalence |
| Yılmaz Tuğan et al. (2022) [45] | 1. Age>18 years<br>2. BCVA of < 20/20 in either eye<br>3. Active or previous uveitis findings<br>4. Pathology on biomicroscopic and fundus examination<br>5. Any retinal and choroidal pathology observed on OCT of the macula and enhanced depth imaging OCT of the choroid | 1. Any other systemic disease | 1. Non-ocular Behcet diagnosed by ISGBD criteria | 1. Without a prior history of ocular or systemic diseases<br>2. Refractive spherical or cylindrical error<2 diopters<br>3. BCVA of ≥ 20/20<br>4. No other ocular or systemic diseases |

Abbreviations: HC, Healthy control; BD. Behcet disease; BU, Behcet uveitis; ISGBD. International Study Group for BD; OCTA, Optical coherence tomography angiography; OCT, Optical coherence tomography; FA, Fluorescein angiography; BCVA, best corrected visual acuity; IOP, Intraocular pressure; AL, Axial length; FAZ, Foveal avascular zone

**Table 3. Retinal layers and regions measured across the included studies.**

| Study | Field of view | Metrics described | Calculation method | Retinal layers measured (acronyms used) | Regions measured | Study Definitions |
|---|---|---|---|---|---|---|
| Accorinti et al. (2019) [24] | Macula: 6×6 mm² | VD | Zeiss | Macula: SCP, DCP, Choriocapillaris | Macula: Whole | – |
| Aksoy et al. (2020) [30] | Macula: 6×6 mm² | VD | AngioVue | Macula: SCP, DCP, Choriocapillaris | Macula: Foveal, Parafoveal (superior, inferior, nasal, temporal), FAZ | **SCP:** Layer thickness of 60 μm from the ILM <br>**DCP:** 30-μm-thick layer from the IPL |
| Balicoglu Yilmaz et al. (2020) [31] | Macula: 6×6 mm² <br>Optic disc: 4.5×4.5 mm² | VD <br>Flow area | AngioVue | Macula: SRCP, Choriocapillaris <br>Optic disk: RPC | Macula: FAZ <br>Optic disk: RPC | – |
| Cheng et al. (2018) [25] | Macula: 3×3 mm² | VD | AngioVue | Macula: SRCP, DRCP | Macula: Whole, Parafoveal (superior, inferior, nasal, temporal) FAZ | **SRCP:** 3 μm below the ILM to 15 μm below the IPL <br>**DRCP:** From 15 to 70 μm below the IPL |
| Comez et al. (2019) [34] | Macula: 6×6 mm² | VD <br>Flow area | AngioVue | Macula: SCP, DCP, Choriocapillaris | Macula: Whole, FAZ | **SCP:** 3mm below the ILM and an outer boundary 16mm below the IPL <br>**DCP:** 16mm to 70mm below IPL |
| Dai et al. (2024) [56] | Macula: 12×12 mm² | VD <br>Flow area | – | Macula: SVP, DCP | Macula: Foveal, Parafoveal, Perifoveal | **SVP:** From the bottom border of the nerve fiber layer to the lower third of the ganglion cell complex <br>**DCP:** From the lower border of ICP to 25 μm below the boundary between INL and OPL |
| Degirmenci et al. (2018) [53] | Macula: - | VD <br>Flow area | AngioVue | Macula: SCP, DCP | Macula: Parafoveal, FAZ | – |
| Emre et al. (2019) [26] | Macula: 6×6 mm² | VD | AngioVue | Macula: SRCP, DRCP, | Macula: Whole FAZ | – |
| Eser-Ozturk et al. (2021) [32] | Macula: 6×6 mm² | VD | IMAGEnet | Macula: SCP, DCP | Macula: Foveal, Parafoveal (superior, inferior, nasal, temporal), FAZ | **SCP:** 2.6mm below the ILM to 15.6mm below the IPL <br>**DCP:** From 15.6 to 70.2mm below the IPL |
| Ferreira et al. (2023) [42] | Macula: 3×3 mm² | VD | Heidelberg | Macula: SVP, DCP, ICP | Macula: Whole, FAZ | **SVP:** The inner half of the IPL and the whole GCL <br>**DCP:** OPL and the outer half of the INL <br>**ICP:** The inner half of the INL, and the outer half of the IPL |
| Goker et al. (2019) [35] | Macula: 6×6 mm² | VD | AngioVue | Macula: SCP, DCP, Choriocapillaris | Macula: Whole (superior and inferior hemisphere), Foveal, Parafoveal, Perifoveal, FAZ | – |
| Guo et al. (2023) [43] | Macula: 6×6 mm² | VD | BM-400K BMizar | Macula: SRV, DRV, Choriocapillaris | Macula: Whole, FAZ | **SRV:** ILM to 9mm below the IPL <br>**DRV:** 6mm below the IPL to 9mm below the ONL <br>**Choriocapillaris:** Bruch membrane to 29mm below the Bruch membrane |
| Kianersi et al. (2024) [49] | Macula: 6×6 mm² <br>Optic disc: 4.5×4.5 mm² | VD | AngioVue | Macula: SRCP, DRCP Choriocapillaris Optic disk: RPC | Macula: Whole, Foveal, Parafoveal (superior, inferior, nasal, temporal), Perifoveal (superior, inferior, nasal, temporal), FAZ <br>Optic disk: RPC (superior, inferior, nasal, temporal) | **SRCP:** Between the posterior margin of the inner layer and the ILM <br>**DRCP:** From the posterior border of the IPL to the posterior margin of the outer layer <br>**RPC:** ILM to the posterior margin of the nerve fiber layer |

*(Continued)*

| Study | Field of view | Metrics described | Calculation method | Retinal layers measured (acronyms used) | Regions measured | Study Definitions |
|---|---|---|---|---|---|---|
| Khairallah et al. (2017) [27] | Macula: 3×3 mm² | VD | ImageJ software | Macula: SCP, DCP | Macula: Whole, FAZ | **SCP:** Between an inner boundary at 5.6 mm beneath the ILM and an outer boundary at 12.6 mm beneath the IPL **DCP:** Between the inner and outer boundaries, respectively, at 15.6 and 70.2 mm beneath the IPL |
| Koca et al. (2019) [37] | Macula: 3×3 mm² | VD | AngioVue | Macula: SCP, DCP | Macula: Whole Parafoveal, FAZ | **SCP:** The upper border of the superficial vascular layer was defined as the ILM offset and the lower border as 9 μm below the IPL **DCP:** 9 μm below the IPL and 9 μm upper the OPL |
| Karalezli et al. (2021) [47] | Macula: 6×6 mm² Optic disc: 4.5×4.5 mm² | VD | AngioAnalytics | Macula: SRCP, DRCP, Choriocapillaris Optic disk: RPC | Macula: Whole, Foveal, Parafoveal, Perifoveal, FAZ Optic disk: RPC | **RPC:** Prolongs from the ILM to the nerve fiber layer |
| Küçük et al. (2022) [48] | Macula: 6×6 mm² Optic disc: 4.5×4.5 mm² | VD | AngioVue | Macula: SCP, DCP, Choriocapillaris Optic disk: RPC | Macula: Whole, Foveal, Parafoveal, Perifoveal, FAZ Optic disk: RPC | – |
| Karaca et al. (2023) [36] | Macula: 6×6 mm² | VD | AngioVue | Macula: SCP, DCP | Macula: Whole Parafoveal, FAZ | – |
| Nassar et al. (2022) [46] | Macula: 3×3 mm² | VD | AngioAnalytics | Macula: SCP, DCP | Macula: Whole, FAZ | **SCP:** 3 mm below the ILM to 15 mm below the IPL **DCP:** 30 μm thick layer from the ILM |
| Pei et al. (2019) [44] | Macula: 3×3 mm² | VD | AngioAnalytics | Macula: SCP, DCP | Macula: Whole, FAZ | **SCP:** ILM to 9 μm above the lower boundary of IPL **DCP:** From 9 μm above the lower boundary of IPL to 9 μm below the lower boundary of OPL |
| Raafat et al. (2019) [38] | Macula: 6×6 mm² | VD | AngioVue | Macula: SCP, DCP | Macula: Whole, FAZ | – |
| Shen et al. (2024) [50] | Macula: 6×6 mm² Optic disc: 4.5×4.5 mm² | VD | AngioVue | Macula: SCP, DCP, Choriocapillaris Optic disk: RPC | Macula: Whole, Parafoveal FAZ Optic disk: RPC | – |
| Smid et al. (2021) [28] | Macula: 3×3 mm² | VD | Heidelberg | Macula: SVP, DCP, ICP | Macula: Parafoveal FAZ | **SVP:** Inner half of the IPL and the whole GCL **DCP:** OPL and the outer half of the INL **ICP:** Inner half of the INL and the outer half of the IPL |
| Simsek et al. (2022) [39] | Macula: 6×6 mm² | VD | ImageJ software | Macula: SRCP, DRCP | Macula: Foveal, Parafoveal, FAZ | – |
| Türkcü et al. (2020) [33] | Macula: 3×3 mm² | VD | AngioVue | Macula: SRCP, DRCP | Macula: Whole, Foveal, Parafoveal (superior, inferior, nasal, temporal), FAZ | – |

*(Continued)*

| Study | Field of view | Metrics described | Calculation method | Retinal layers measured (acronyms used) | Regions measured | Study Definitions |
|---|---|---|---|---|---|---|
| Yan et al. (2021) [29] | Macula: 6×6 mm² Optic disc: 4.5×4.5 mm² | VD | AngioVue | Macula: SCP, DCP Choriocapillaris Optic disk: RPC | Macula: Whole, Foveal, Parafoveal (superior, inferior, nasal, temporal), Perifoveal (superior, inferior, nasal, temporal), FAZ Optic disk: RPC (superior, inferior, nasal, temporal) | **FAZ:** Extending from the ILM to the OPL **SCP:** ILM to IPL **DCP:** IPL to OPL **RPC:** ILM and nerve fiber layer **Choriocapillaris:** Bruch membrane to Bruch membrane +30μm |
| Yilmaz et al. (2021) [40] | Macula: 6×6 mm² | VD | AngioAnalytics | Macula: SCP, DCP | Macula: Foveal, Parafoveal, Perifoveal, FAZ | – |
| Yılmaz Tuğan et al. (2022) [45] | Macula: 6×6 mm² Optic disc: 4.5×4.5 mm² | VD | AngioVue | Macula: SRCP, DRCP, Choriocapillaris Optic disk: RPC | Macula: Whole, Foveal, Parafoveal (superior, inferior, nasal, temporal), Perifoveal (superior, inferior, nasal, temporal) FAZ Optic disk: RPC | **SCP:** ILM to IPL **DCP:** IPL to OPL **Choriocapillaris:** Bruch membrane to 30μm inferiorly to the Bruch membrane |

*Abbreviations:* VD, Vessel density; SRCP, Superficial retinal capillary plexus; SVP, Superficial vascular plexus; SCP, Superficial capillary plexus; Superficial retinal layer; DRCP, Deep retinal capillary plexus; DCP, Deep capillary plexus; DRL, Deep retinal layer; ICP, Intermediate capillary plexus; FAZ, Foveal avascular zone; RPC, Radial peripapillary capillary; RPC, Radial peripapillary capillary; ILM, Internal limiting membrane; IPL, Inner plexiform layer; OPL, Outer plexiform layer; INL, Inner nuclear layer; ONL, Outer nuclear layer; GCL, Ganglion cell layer

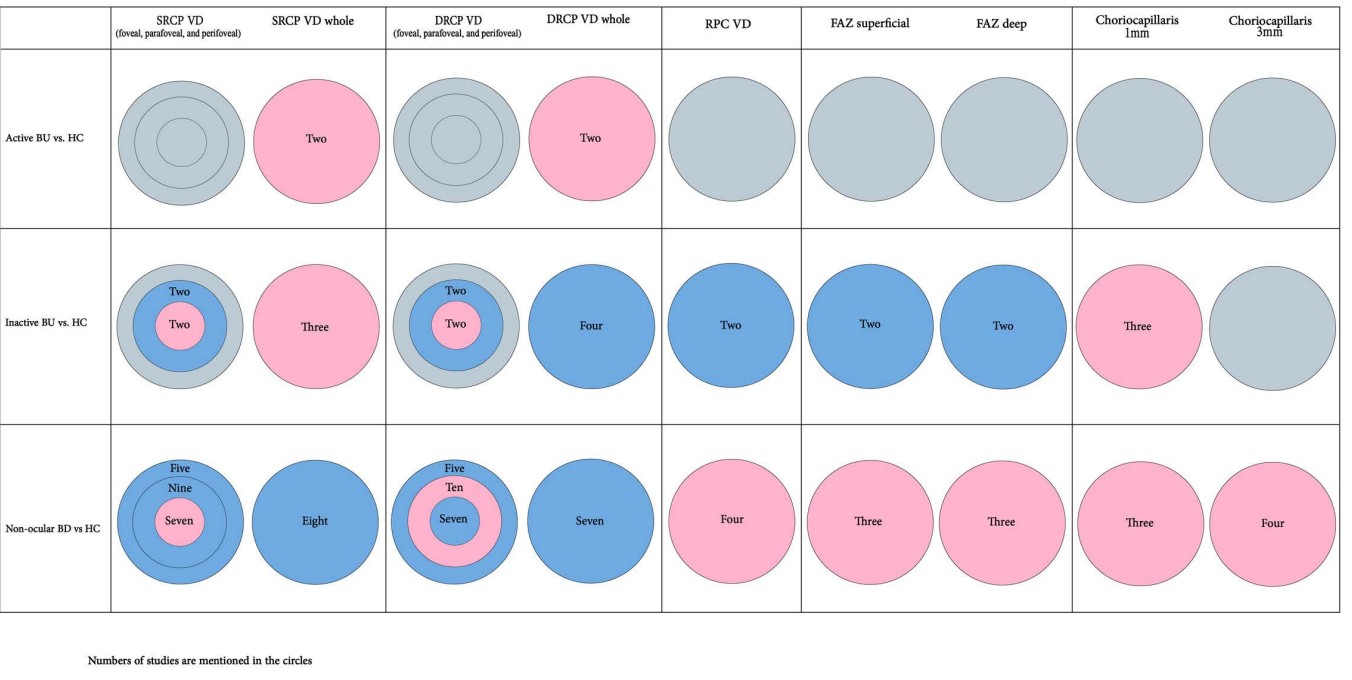

**Fig 2. Summary of the main meta-analyses of the study regarding VD.** Abbreviations: BU: Bechet's uveitis, DRCP: deep retinal capillary plexus, HC: healthy control, RPC: retinal peripapillary capillary, SRCP: superficial retinal capillary plexus, VD: vessel density.

**Table 4. Summary of the meta-analysis results.**

| Outcome | | Comparison | No. of studies | Hedges g | CI | I² | Unadjusted p-value | Corrected p-value |
|---|---|---|---|---|---|---|---|---|
| SRCP | Whole (6*6) VD | Active BU vs. HC | 2 | −0.59 | −2.07 to 0.88 | 74.28% | 0.43 | 0.43 |
| | | Non-Ocular BD vs. HC | 8 | −0.61 | −1.11 to −0.12 | 88.08% | **0.01** | **0.02** |
| | Whole (3*3) VD | Inactive BU vs. HC | 4 | −0.75 | −6.53 to 1.03 | 98.64% | 0.15 | – |
| | Foveal VD | Inactive BU vs. HC | 2 | −0.24 | −0.70 to 0.23 | 43.80% | 0.32 | 0.46 |
| | | Non-Ocular BD vs. HC | 7 | −0.14 | −0.31 to 0.04 | 0.59% | 0.13 | 0.39 |
| | | Non-specified BD vs. HC | 2 | −0.12 | −0.45 to 0.21 | 0% | 0.46 | 0.46 |
| | Parafoveal VD | Inactive BU vs. HC | 2 | −1.10 | −1.80 to −0.41 | 74.30% | **0.001** | **0.003** |
| | | Non-Ocular BD vs. HC | 8 | −0.21 | −0.36 to −0.06 | 14.62% | **0.005** | **0.005** |
| | | Non-specified BD vs. HC | 3 | −0.75 | -.1153 to −0.34 | 56.01% | **0.003** | **0.004** |
| | Perifoveal VD | Non-Ocular BD vs. HC | 5 | −0.28 | −0.5 to −0.06 | 0.% | **0.01** | **0.01** |
| | | Non-specified BD vs. HC | 2 | −0.44 | −0.78 to −0.11 | 0% | **0.00001** | **0.00002** |
| DRCP | Whole (6*6) VD | Active BU vs. HC | 2 | −0.36 | −1.12 to 0.40 | 74.28% | 0.36 | 0.36 |
| | | Non-Ocular BD vs. HC | 7 | −0.6 | −1.03 to −0.17 | 83.33% | **0.006** | **0.012** |
| | Whole (3*3) VD | Inactive BU vs. HC | 4 | −1.32 | −1.68 to −0.95 | 47.31% | 0.00001 | – |
| | Foveal VD | Inactive BU vs. HC | 2 | −0.37 | −0.85 to 0.10 | 45.84% | 0.13 | 0.13 |
| | | Non-Ocular BD vs. HC | 7 | −0.7 | −1.21 to −0.18 | 86.58% | **0.007** | **0.012** |
| | | Non-specified BD vs. HC | 2 | −0.44 | −0.78 to −0.11 | 0% | **0.008** | **0.012** |
| | Parafoveal VD | Inactive BU vs. HC | 2 | −1.10 | −1.80 to −0.41 | 74.30% | **0.00001** | **0.000015** |
| | | Non-Ocular BD vs. HC | 10 | −0.27 | −0.56 to 0.02 | 77.36% | 0.06 | 0.06 |
| | | Non-specified BD vs. HC | 3 | −0.81 | −1.07 to −0.54 | 0% | **0.00001** | **0.000015** |
| | Perifoveal VD | Non-Ocular BD vs. HC | 5 | −0.45 | −0.77 to −0.12 | 49.75% | **0.007** | **0.014** |
| | | Non-specified BD vs. HC | 2 | 0.38 | −0.08 to 0.84 | 44.64% | 0.1 | 0.1 |
| RPC VD | | Inactive BU vs. HC | 2 | −1.22 | −1.66 to −0.78 | 0% | **0.00001** | **0.00002** |
| | | Non-Ocular BD vs. HC | 4 | −0.48 | −0.99 to 0.03 | 79.41% | 0.06 | 0.06 |
| Choriocapillaris flow area | | Inactive BU vs. HC | 2 | −0.20 | −0.51 to 0.11 | 0% | 0.22 | 0.66 |
| | | Non-Ocular BD vs. HC (1 mm) | 3 | 0.13 | −0.58 to 0.32 | 60.84% | 0.58 | 0.75 |
| | | Non-Ocular BD vs. HC (3 mm) | 4 | 0.10 | −0.51 to 0.70 | 84.75% | 0.75 | 0.75 |
| FAZ | FAZ | Active BU vs. HC | 2 | 0.853 | −0.05 to 1.76 | 77.39% | 0.06 | 0.06 |
| | | Inactive BU vs. HC | 4 | 1.22 | −0.07 to 2.50 | 96.37% | 0.06 | 0.06 |
| | | Non-Ocular BD vs. HC | 10 | 0.82 | 0.01 to 1.63 | 96.12% | 0.05 | 0.06 |
| | | Non-specified BD vs. HC | 2 | −0.92 | −1.62 to −0.21 | 72.64% | **0.01** | **0.04** |
| | FAZ Superficial | Inactive BU vs. HC | 5 | 0.27 | 0.05 to 0.48 | 0% | **0.01** | **0.02** |
| | | Non-Ocular BD vs. HC | 6 | 0.46 | 0.03 to 0.90 | 77.26% | **0.04** | **0.04** |
| | FAZ Deep | Inactive BU vs. HC | 4 | 0.48 | 0.08 to 0.87 | 59.18% | **0.02** | **0.04** |
| | | Non-Ocular BD vs. HC | 3 | 0.45 | −0.11 to 1.02 | 80.02% | 0.12 | 0.12 |

Corrected p-values were calculated using the FDR method, with significant values boldfaced. Abbreviations: BD, Behcet Disease; BU, Behcet Uveitis; HC, Healthy Controls; CI, Confidence Interval; FDR, False discovery rate; VD, Vessel density; SRCP, Superficial retinal capillary plexus; DRCP, Deep retinal capillary plexus; FAZ, Foveal avascular zone; RPC, Radial peripapillary capillary.

difference between the two groups (Hedges g = −0.75, CI= [−6.53 to 1.03], I² = 98.64%, P value = 0.15) (Fig 4A). Values for the whole SRCP VD in the study by Pei et al. [44] were excluded from this analysis, since they were not provided in mean and SD and also were not convertible to this form due to being skewed away from normality.

**3.4.2. Foveal SRCP VD** (inactive BU vs. HC). Two studies compared SRCP foveal VD between inactive BU and HC subgroups using the RTVue XR Avanti device [30, 33]. Meta-analysis (65 inactive BU and 61 HC eyes) showed no significant

Table 5. The direction of effects regarding comparisons of OCTA parameters across included studies.

| Study (year) | FAZ area | FAZ sup | FAZ deep | Superficial whole VD | Superficial foveal VD | Superficial parafoveal VD | Superficial perifoveal VD | Deep Whole VD | Deep foveal VD | Deep parafoveal VD | Deep Perifoveal VD | Chorio-capillaris | RPC |
|---|---|---|---|---|---|---|---|---|---|---|---|---|---|
| Accorinti et al. (2019) [24] | — | — | — | ↓* ~** ~c | — | — | — | ↓* ~** ~c | — | — | — | ~* ~** ~c | — |
| Aksoy et al. (2020) [30] | — | ↑** | ↑** | — | ~** | ↓** (inferior and nasal subregions) | — | — | ↓** | ↓** (superior, inferior, nasal and temporal subregions) | — | ~** | — |
| Balicoglu Yilmaz et al. (2020) [31] | — | ~** | — | — | — | — | — | — | — | — | — | ~** | ↓** |
| Cheng et al. (2018) [25] | — | ~** | ~** | ↓** | — | ↓** (superior, inferior, nasal and temporal subregions) | — | ↓** | — | ↓** (superior, inferior, nasal and temporal subregions) | — | — | — |
| Comez et al. (2019) [34] | — | ~***** | ↓***** ~ | ↓**** | — | — | — | ↓**** | — | — | — | ↓**** | — |
| Dai et al. (2024) [56] | — | — | — | — | ↓** | ↓** (1–6mm) | ~** (6–12 mm) | — | ↓** | ↓** (1–6 mm) | ↓** (6–12 mm) | — | — |
| Degirmenci et al. (2018) [53] | — | ↑***** | ↑***** | — | — | ~**** | — | — | — | ~**** | — | — | — |
| Emre et al. (2019) [26] | — | — | — | ~* | — | — | — | ↓* | — | — | — | — | — |
| Eser-Ozturk et al. (2021) [32] | — | ~** | ~** | — | ~** | ↓** (superior, inferior, nasal and temporal subregions) | — | — | ~** | ↓** (superior, inferior, nasal and temporal subregions) | — | — | — |
| Ferreira et al. (2023) [42] | ~** ~**** ~b | — | — | ↓** (nasal subregion) ~**** ~b | — | — | — | ↓** (superior, inferior, nasal and temporal subregions) ~**** ↓b (superior, inferior, nasal and temporal subregions) | — | — | — | — | — |
| Goker et al. (2019) [35] | ↑**** ↑***** | — | ~***** | ↓**** | ~**** | ~**** | ~**** | ↓**** | ~**** | ~**** | ~**** | — | |
| Guo et al. (2023) [43] | — | ↑** | ↑** | ~** | — | — | — | ~** | — | — | — | — | — |

(Continued)

Table 5. (Continued)

| Study (year) | FAZ area | FAZ sup | FAZ deep | Superficial whole VD | Superficial foveal VD | Superficial parafoveal VD | Superficial perifoveal VD | Deep Whole VD | Deep foveal VD | Deep parafoveal VD | Deep Perifoveal VD | Chorio-capillaris | RPC |
|---|---|---|---|---|---|---|---|---|---|---|---|---|---|
| Kianersi et al. (2024) [49] | ~***  ↑**** | — | — | ↓***  ~**** | ↓***  ~**** | ↓*** (superior, nasal inferior and temporal subregions) ↓**** (superior, inferior, nasal and temporal subregions) | ↓*** (superior, inferior, nasal and temporal subregions) ~**** (except for temporal subregion) | ↓***  ↓**** | ↓***  ↓**** | ↓*** (superior, nasal inferior and temporal subregions) ↓**** (superior, inferior, nasal and temporal subregions) | ↓*** (superior, inferior, nasal and temporal subregions) ↓**** (superior, inferior, nasal and temporal subregions) | ~***  ↑**** | ↓***  ↓**** |
| Khairallah et al. (2017) [27] | — | ~* | ~* | ~* | — | — | — | ↓* | — | — | — | — | — |
| Koca et al. (2019) [37] | ~***  ~**** | — | — | ↓***  ~**** | — | ↓***  ~**** | — | ↓***  ~**** | — | ↓***  ~**** | — | — | — |
| Karalezli et al. (2021) [47] | ↑**** | — | — | ↓****  ~ | ~****  ~ | ~**** | — | ↓**** | ↓****  ~ | ↓**** | — | ~**** | ↓**** |
| Küçük et al. (2022) [48] | ~**** | — | — | ~**** | ~**** | ~**** | ~**** | ↓**** | ~**** | ↓**** | ↓**** | ~**** | ~**** |
| Karaca et al. (2023) [36] | ↑**  ~****  ~b | — | — | ↓**  ~****  ↓b | — | ↓**  ~****  ↓b | — | ↓**  ~****  ↓b | — | ↓**  ~****  ↓b | — | — | — |
| Nassar et al. (2022) [46] | ↑** | — | — | ↓** | — | — | — | ↓** | — | — | — | — | — |
| Pei et al. (2019) [44] | ↑** | — | — | ↓** | — | — | — | ↓** | — | — | — | — | — |
| Raafat et al. (2019) [38] | — | ~**** | — | ↓**** | — | — | — | — | — | — | — | — | — |
| Shen et al. (2024) [50] | ~* | — | — | ↓* | — | ↓* (1–6 mm) | — | ↓* | — | ↓* (1–6 mm) | — | ↓* | ↓* |
| Smid et al. (2021) [28] | ↓*  ↑****  ←a | — | — | — | — | ↓*  ~****  ↓a | — | — | — | ↓*  ~****  ↓a | — | — | — |
| Simsek et al. (2022) [39] | — | ~**** | ~**** | — | ~**** | ~**** | — | — | ~**** | ~**** | — | — | — |
| Türkcü et al. (2020) [33] | — | ~** | ↑** | ↓** | ↓** | ↓** (superior, inferior, nasal and temporal subregions) | — | ↓** | ↓** | ↓** (superior, inferior, nasal and temporal subregions) | — | — | — |

*(Continued)*

| Study (year) | FAZ area | FAZ sup | FAZ deep | Superficial whole VD | Superficial foveal VD | Superficial parafoveal VD | Superficial perifoveal VD | Deep Whole VD | Deep foveal VD | Deep parafoveal VD | Deep Perifoveal VD | Chorio-capillaris | RPC |
|---|---|---|---|---|---|---|---|---|---|---|---|---|---|
| Yan et al. (2021) [29] | ~* | – | – | ~* | ~* | ~* (superior, inferior, nasal, and temporal subregions) | ~* (superior, inferior, nasal, and temporal subregions) | ~* | ~* | ~* (except for ↓* In the nasal subregion) | ~* (superior, inferior, nasal, and temporal subregions) | ↓* | ↓* (in every sub-region except for tem-poral) |
| Yılmaz et al. (2021) [40] | ~*** ~**** | – | – | – | ~*** ~**** | ↓*** ~**** | ↓*** ~**** | – | ~*** ~**** | ↓*** ~**** | ↓*** ~**** | – | – |
| Yılmaz Tuğan et al. (2022) [45] | ~**** ~**** | ~**** | – | ~**** | ~**** | ~**** | ~**** | ↓**** | ↓**** → | ↓**** → | ↓**** → | ~**** | ~**** |

-: Data not available

*: Active BU compared to HC

**: Inactive BU compared to HC

***: Non-specified ocular BD compared to HC

****: Non-ocular BD compared to HC

a : Active BU compared to non-ocular BD

b : Inactive BU compared to non-ocular BD

C : Active BU compared to inactive BU

↓: reduced

↑: increased

~: no significant difference

*Abbreviations:* BU, Behcet uveitis; BD, Behcet disease; HC, Healthy control; OCTA, Optical coherence tomography angiography; FAZ, Foveal avascular zone; VD, Vessel density; RPC, Radial peripapillary capillary

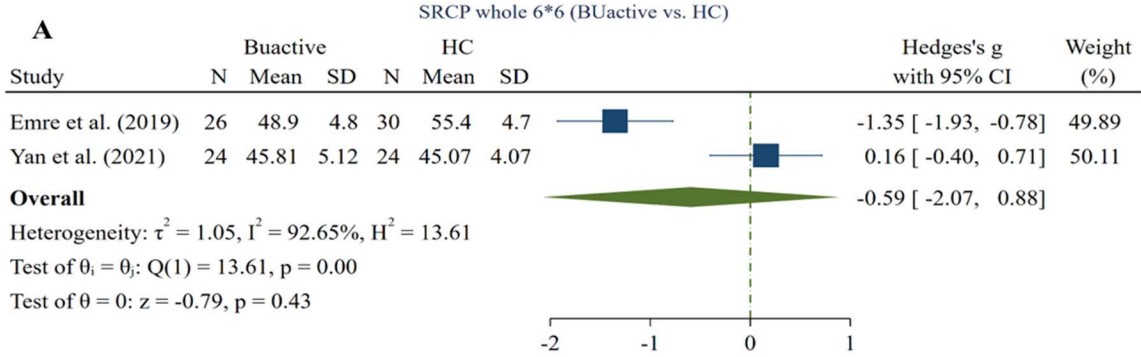

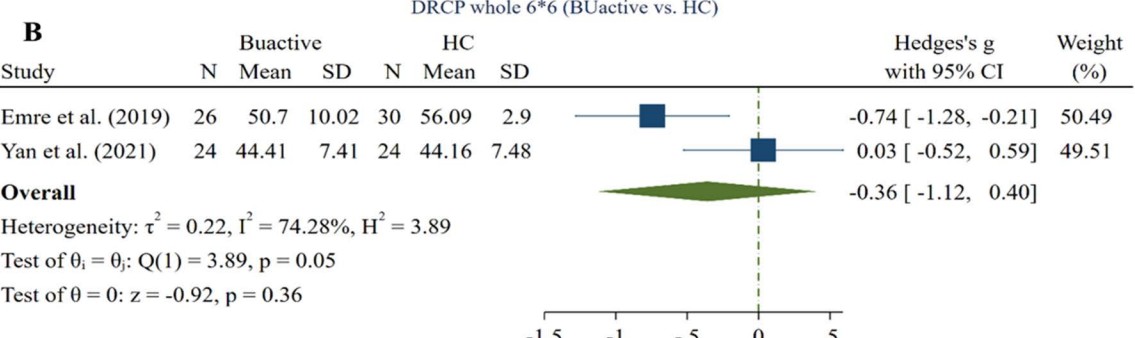

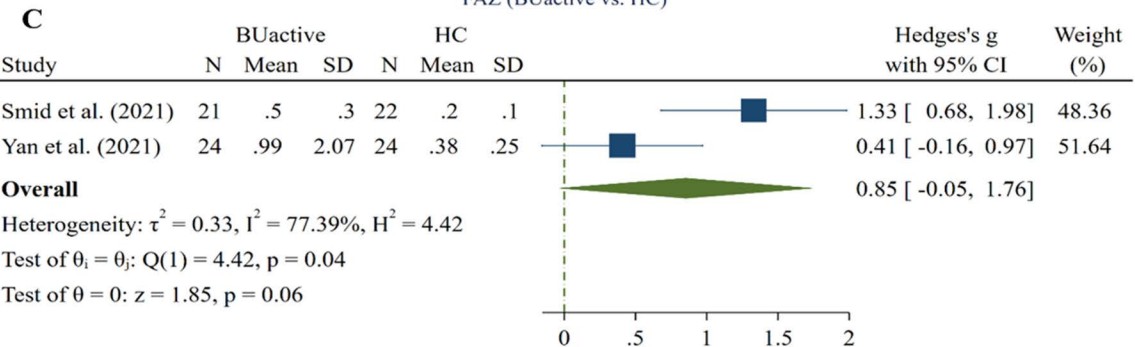

**Fig 3. Active BU eyes vs. HCs.** A: whole image (3*3) SRCP VD in active BU and HC eyes, B: whole image (3*3) DRCP VD in active BU and HC eyes, C: FAZ in active BU and HC eyes. Abbreviations: BU: Bechet's uveitis, CI: confidence interval, DRCP: deep retinal capillary plexus, SRCP: superficial retinal capillary plexus, SD: standard deviation, VD: vessel density.

difference between the two groups (Hedges g = −0.24, CI= [−0.70 to 0.23], $I^2$ = 43.80%, P value = 0.32, Corrected P value = 0.46) (**Fig 4B**).

 **3.4.3. Parafoveal SRCP VD** (inactive BU vs. HC). Two studies compared SRCP parafoveal VD between inactive BU and HC subgroups using the RTVue XR Avanti device [30, 36]. Meta-analysis (72 inactive BU and 74 HC eyes) showed a significant reduction in the active BU group compared to the HC (Hedges g = −1.10, CI= [−1.80 to −0.41], $I^2$ = 74.30%, P value = 0.001, Corrected P value = 0.003) (**Fig 4C**).

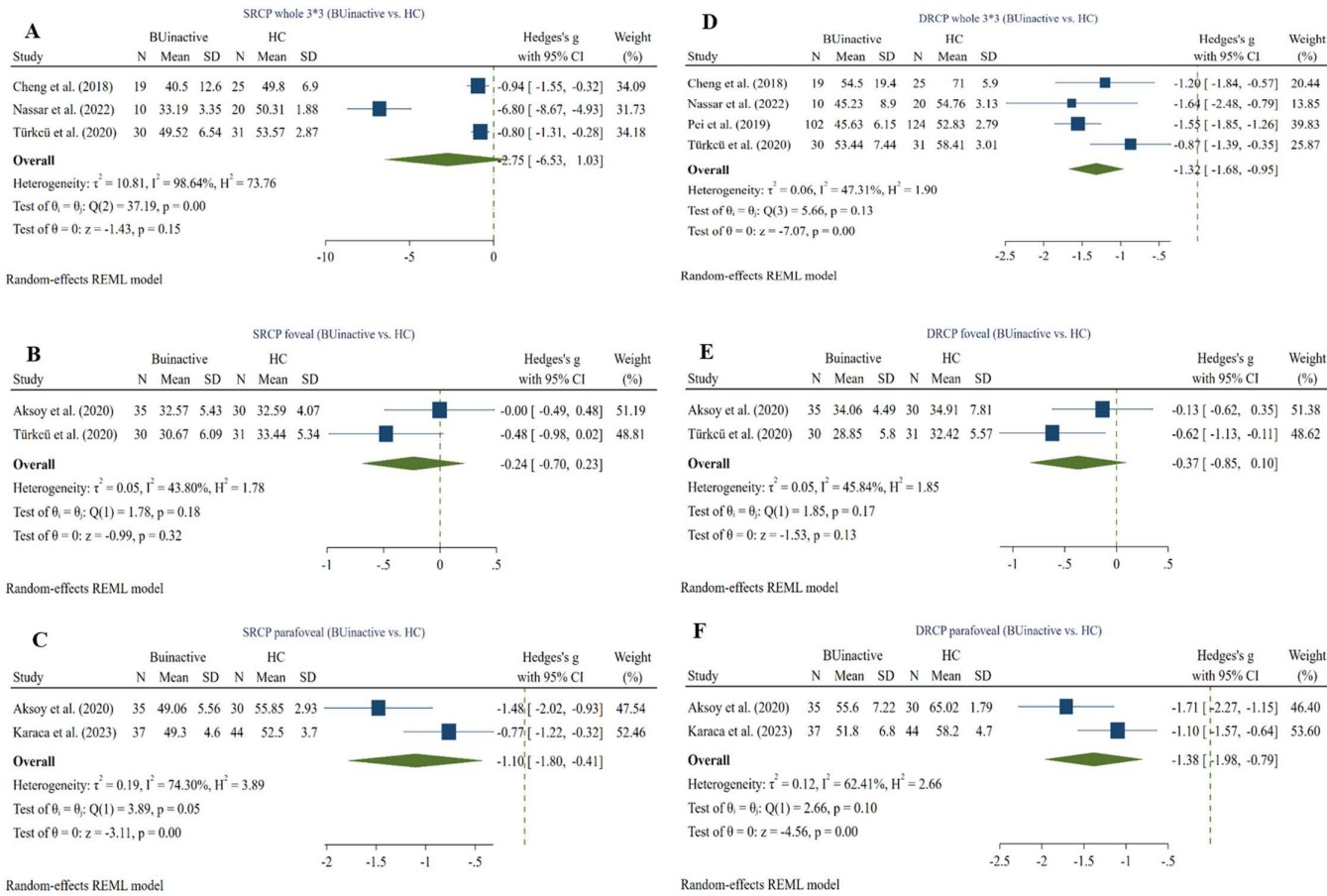

**Fig 4. Inactive BU eyes vs. HCs.** A: whole image (3*3) SRCP VD in inactive BU and HC eyes, B: foveal SRCP in inactive BU and HC eyes, C: parafoveal SRCP in inactive BU and HC eyes, D: whole image (3*3) DRCP VD in inactive BU and HC eyes, E: foveal DRCP in inactive BU and HC eyes, F: parafoveal DRCP in inactive BU and HC eyes. *Abbreviations:* BU: Bechet's uveitis, CI: confidence interval, DRCP: deep retinal capillary plexus, SRCP: superficial retinal capillary plexus, SD: standard deviation, VD: vessel density.

**3.4.4. DRCP whole (3*3) VD** (inactive BU vs. HC). Four studies compared DRCP whole (3*3) VD between inactive BU and HC subgroups using the RTVue XR Avanti device [25,33,44,46,57]. Meta-analysis (161 inactive BU and 200 HC eyes) revealed a significant reduction in the inactive BU group (Hedges g = −1.32, CI= [−1.68 to −0.95], $I^2$ = 47.31%, P value = 0.00001) (**Fig 4D**).

**3.4.5. Foveal DRCP VD** (inactive BU vs. HC). Two studies compared DRCP foveal VD between inactive BU and HC subgroups using the RTVue XR Avanti device [30, 33]. Meta-analysis (65 inactive BU and 61 HC eyes) revealed no significant difference between the groups (Hedges g = −0.37, CI= [−0.85 to 0.10], $I^2$ = 45.84%, P value = 0.13, Corrected P value = 0.13) (**Fig 4E**).

**3.4.6. Parafoveal DRCP VD** (inactive BU vs. HC). Two studies compared DRCP parafoveal VD between inactive BU and HC subgroups using the RTVue XR Avanti device [30, 36]. Meta-analysis (72 inactive BU and 74 HC eyes) revealed a significant reduction in the active BU group (Hedges g = −1.38, CI= [−1.97 to −0.78], $I^2$ = 62.41%, P value = 0.00001, Corrected P value = 0.000015) (**Fig 4F**).

**3.4.7. RPC VD** (inactive BU vs. HC). One study evaluating both right and left eyes compared RPC VD between inactive BU and HC subgroups using the RTVue XR Avanti device [31]. Meta-analysis of the left and right eyes' data (40 inactive

BU and 52 HC eyes) revealed a significant reduction in the inactive BU group (Hedges g = −1.22, CI= [−1.66 to −0.78], $I^2$=0.00%, P value = 0.00001, Corrected P value = 0.00002) (**Fig 5A**).

**3.4.8. Choriocapillaris flow area** (inactive BU vs. HC). Two studies, one assessing both eyes, compared the Choriocapillaris flow area (1 mm) between inactive BU and HC subgroups using the RTVue XR Avanti device [30, 31]. Meta-analysis (75 inactive BU and 82 HC eyes) revealed no significant difference between the two groups (Hedges g = −0.20, CI= [−0.51 to 0.11], $I^2$=0.00%, P value = 0.22, Corrected P value = 0.66) (**Fig 5B**).

**3.4.9. FAZ** (inactive BU vs. HC). Four studies compared FAZ between inactive BU and HC subgroups utilizing the RTVue XR Avanti [36, 44, 46] and Heidelberg devices [42]. Meta-analysis (172 inactive BU and 214 HC eyes) revealed no significant difference between the two groups (Hedges g = 1.22, CI= [−0.07 to 2.50], $I^2$=96.37%, P value = 0.06, Corrected P value = 0.06) (**Fig 5C**). However, conducting a leave-one-out analysis (removing Mostafa et al. study [46]) revealed a significant difference between the two groups (P value <0.001).

**3.4.10. FAZ superficial** (inactive BU vs. HC). Six studies compared superficial FAZ between inactive BU and HC subgroups applying RTVue XR Avanti [30, 31, 33, 46], Heidelberg [25], and Topcon [32] devices. Meta-analysis (166 inactive BU and

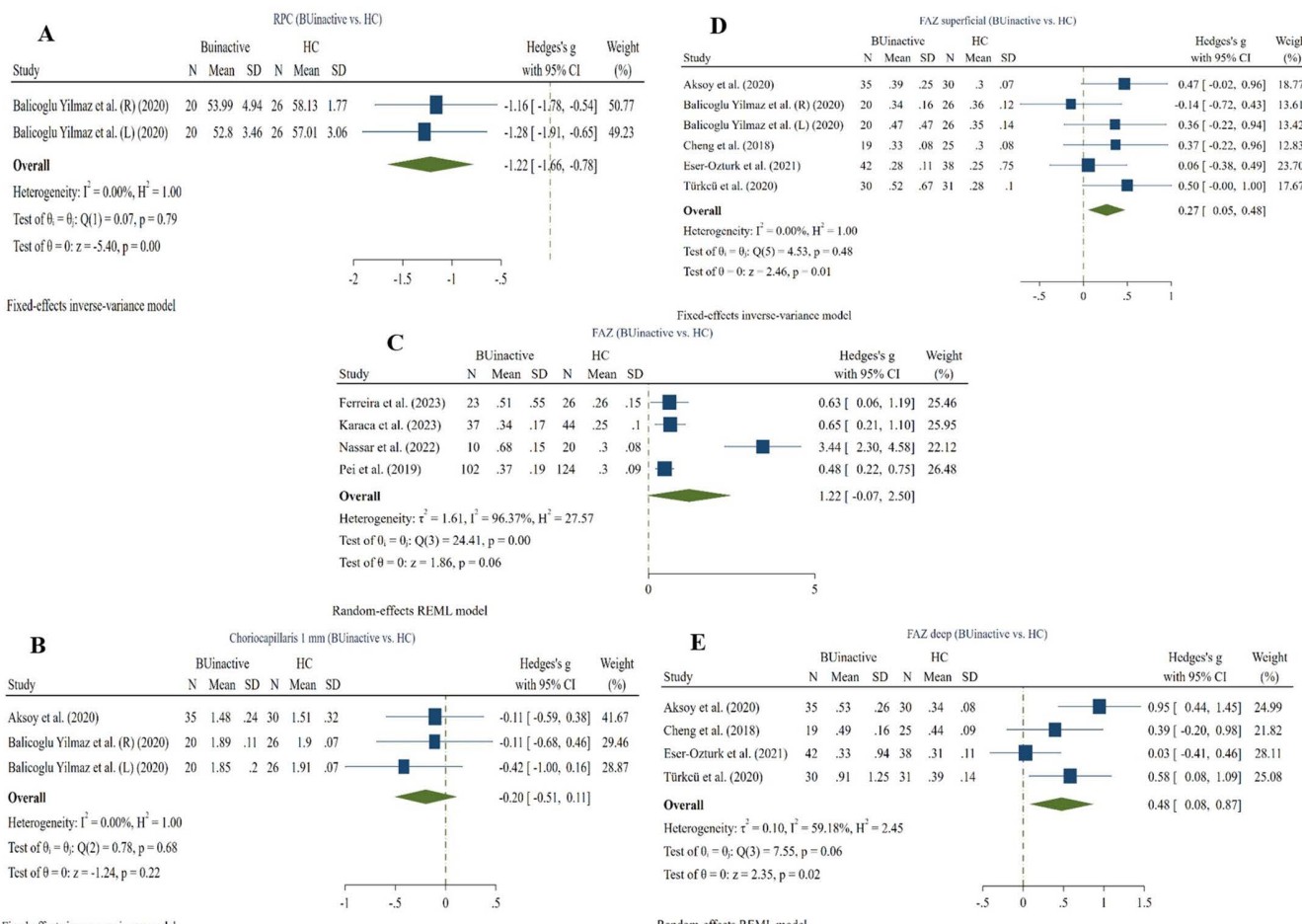

**Fig 5. Inactive BU eyes vs. HCs.** A: RPC VD in inactive BU and HC eyes, B: Choriocapillaris flow area (1 mm) in inactive BU and HC eyes, C: FAZ in inactive BU and HC eyes, D: Superficial FAZ in inactive BU and HC eyes, E: Deep FAZ in inactive BU and HC eyes. Abbreviations: BU: Bechet's uveitis, CI: confidence interval, FAZ: foveal avascular zone, RPC: radial peripapillary capillary, VD: vessel density.

176 HC eyes) revealed a significant difference between the two groups (Hedges g = 0.27, CI= [0.05 to 0.48], I² = 0.00%, P value = 0.01, Corrected P value = 0.02) (**Fig 5D**).

**3.4.11. FAZ deep** (inactive BU vs. HC). Four studies compared deep FAZ between inactive BU and HC subgroups utilizing the RTVue XR Avanti [30 33], Heidelberg [25], and Topcon [32] devices. Meta-analysis of the four studies (126 inactive BU and 124 HC eyes) revealed a significant difference between the two groups, as inactive BU had greater metric compared to the HC (Hedges g = 0.48, CI= [0.08 to 0.87], I² = 59.18%, P value = 0.02, Corrected P value = 0.04) (**Fig 5E**).

## 3.5. Non-ocular BD vs. HC

**3.5.1. Whole (6*6) SRCP VD** (non-ocular BD vs. HC). Nine studies compared SRCP whole (6*6) VD between non-ocular BD and HC subgroups utilizing the RTVue XR Avanti device [24, 34–36, 38, 45, 47–49]. Meta-analysis (284 non-ocular BD and 310 HC eyes) revealed a significant reduction in the non-ocular BD group (Hedges g = −0.61, CI= [−1.11 to −0.12], I² = 88.08%, P value = 0.01, Corrected P value = 0.02) (**Fig 6A**).

**3.5.2. Foveal SRCP VD** (non-ocular BD vs. HC). Seven studies compared SRCP foveal VD between non-ocular BD and HC subgroups applying the RTVue XR Avanti device [35, 39, 40, 45, 47–49]. Meta-analysis (232 non-ocular BD and 271 HC eyes) revealed no statistically significant difference between the groups (Hedges g = −0.14, CI= [−0.31 to 0.04], I² = 0.59%, P value = 0.13, Corrected P value = 0.39) (**Fig 6B**).

**3.5.3. Parafoveal SRCP VD** (non-ocular BD vs. HC). Eight studies compared SRCP parafoveal VD between non-ocular BD and HC subgroups utilizing the RTVue XR Avanti device [35–37, 39, 40, 45, 47, 48]. Meta-analysis (331 non-ocular BD and 368 HC eyes) demonstrated a significant increase in favor of the HC group (Hedges g = −0.21, CI= [−0.36 to −0.06], I² = 14.62%, P value = 0.005, Corrected P value = 0.005) (**Fig 6C**).

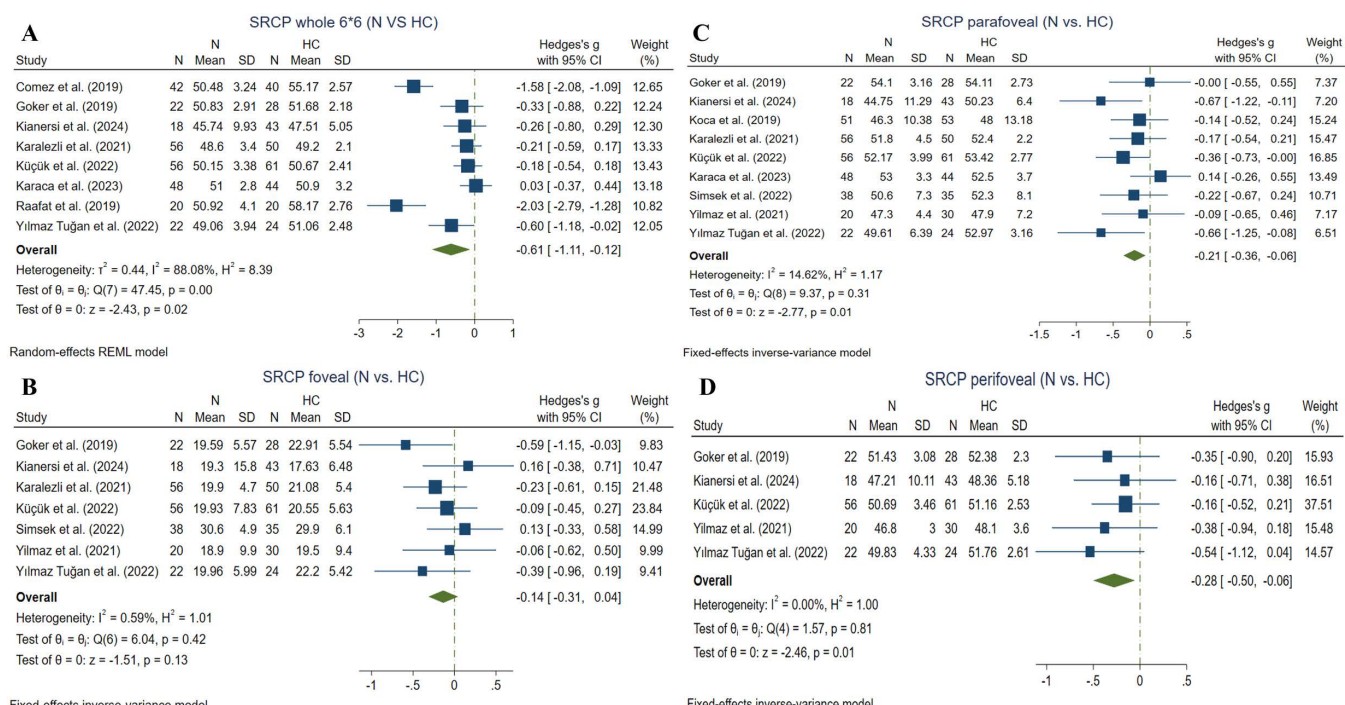

**Fig 6. Non-ocular BD eyes vs. HCs.** A: whole image (6*6) SRCP VD in non-ocular BD and HC eyes, B: Foveal SRCP VD in non-ocular BD and HC eyes, C: Parafoveal SRCP VD in non-ocular BD and HC eyes, D: Perifoveal SRCP VD in non-ocular BD and HC eyes. Abbreviations: BD: Bechet disease, CI: confidence interval, N: non-ocular BD, SRCP: superficial retinal capillary plexus, SD: standard deviation, VD: vessel density.

**3.5.4. Perifoveal SRCP VD** (non-ocular BD vs. HC). Five studies compared SRCP perifoveal VD between non-ocular BD and HC subgroups utilizing the RTVue XR Avanti device [35, 40, 45, 48, 49]. Meta-analysis (138 non-ocular BD and 186 HC eyes) demonstrated significantly greater value in favor of the HC group (Hedges g = −0.28, CI= [−0.5 to −0.06], I² = 0.00%, P value = 0.01, Corrected P value = 0.01) (Fig 6D).

**3.5.5. Whole (6*6) DRCP VD** (non-ocular BD vs. HC). Seven studies compared DRCP whole (6*6) VD between non-ocular BD and HC subgroups utilizing the RTVue XR Avanti device [34–36, 45, 47–49]. Meta-analysis of the studies (259 non-ocular BD and 300 HC eyes) revealed a significantly decreased value among non-ocular BD than HC (Hedges g = −0.6, CI= [−1.03 to −0.17], I² = 83.33%, P value = 0.006, Corrected P value = 0.012) (Fig 7A).

**3.5.6. Foveal DRCP VD** (non-ocular BD vs. HC). Seven studies compared DRCP foveal VD between non-ocular BD and HC subgroups utilizing the RTVue XR Avanti device [35, 39, 40, 45, 47–49]. Meta-analysis (232 non-ocular BD and 271 HC eyes) revealed significantly greater metric for HC than patients (Hedges g = −0.7, CI= [−1.21 to −0.18], I² = 86.58%, P value = 0.007, Corrected P value = 0.012) (Fig 7B).

**3.5.7. Parafoveal DRCP VD** (non-ocular BD vs. HC). Ten studies compared DRCP parafoveal VD between non-ocular BD and HC subgroups utilizing the RTVue XR Avanti device [35–37, 39, 40, 45, 47–49, 53]. Meta-analysis (375 non-ocular BD and 417 HC eyes) revealed no significant difference between the two groups (Hedges g = −0.27, CI= [−0.56 to 0.02], I² = 77.36%, P value = 0.06, Corrected P value = 0.06) (Fig 7C).

**3.5.8. Perifoveal DRCP VD** (non-ocular BD vs. HC). Five studies compared DRCP perifoveal VD between non-ocular BD and HC subgroups utilizing the RTVue XR Avanti device [35, 40, 45, 48, 49]. Meta-analysis (138 non-ocular BD and 186 HC eyes) revealed a significant reduction in the non-ocular BD eyes (Hedges g = −0.45, CI = [−0.77 to −0.12], I² = 49.75%, P value = 0.007, Corrected P value = 0.014) (Fig 7D).

**3.5.9. RPC VD** (non-ocular BD vs. HC). Four studies compared RPC VD between non-ocular BD and HC subgroups utilizing the RTVue XR Avanti device [45, 47–49]. Meta-analysis (152 non-ocular BD and 178 HC eyes) showed no significant

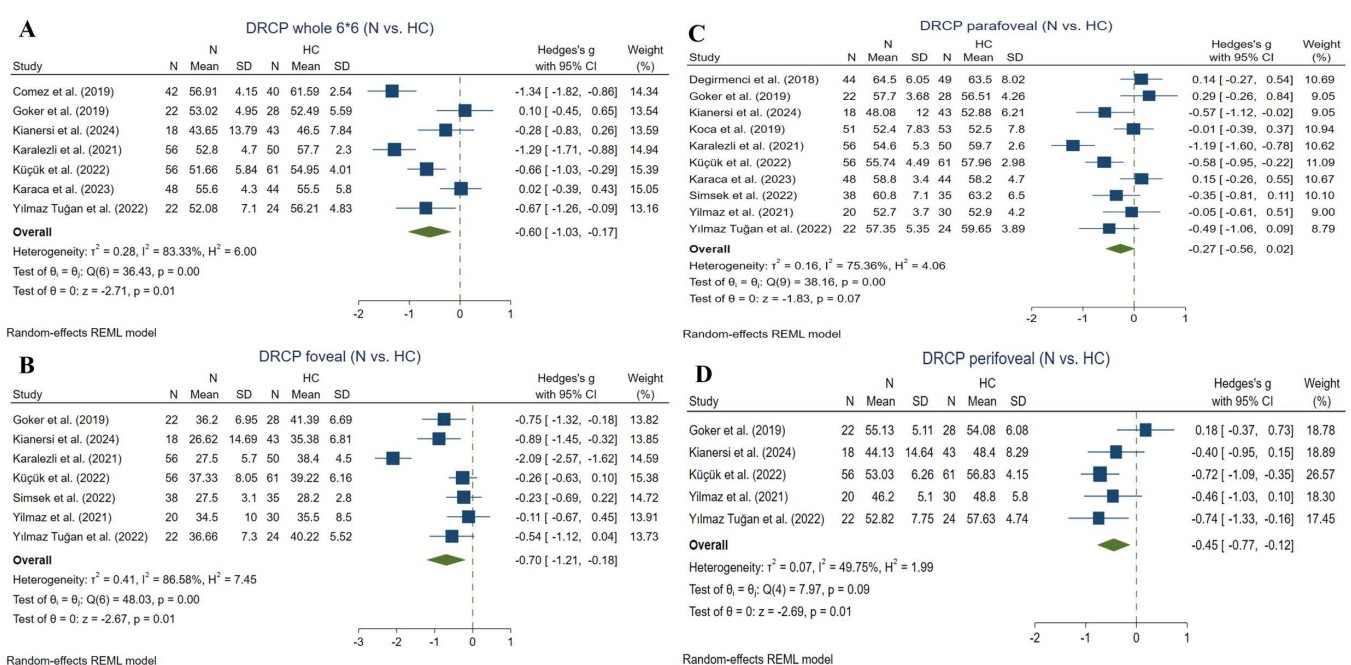

**Fig 7. Non-ocular BD eyes vs. HCs.** A: whole image (6*6) DRCP VD in non-ocular BD and HC eyes, B: Foveal DRCP VD in non-ocular BU and HC eyes, C: Parafoveal DRCP VD in non-ocular BD and HC eyes, D: Perifoveal DRCP VD in non-ocular BD and HC eyes. Abbreviations: BD: Bechet disease, CI: confidence interval, DRCP: deep retinal capillary plexus, N: non-ocular BD, SD: standard deviation, VD: vessel density.

difference between the two groups (Hedges g = −0.48, CI = [−0.99 to 0.03], I²=79.41%, P value=0.06, Corrected P value=0.06) (Fig 8A).

   **3.5.10. Choriocapillaris flow area** (non-ocular BD vs. HC). Three studies compared choriocapillaris flow area (1 mm) between non-ocular BD and HC subgroups utilizing the RTVue XR Avanti device [35, 45, 48]. Meta-analysis (100 non-ocular BD and 113 HC eyes) demonstrated no significant difference between the two groups (Hedges g=0.13, CI = [−0.58 to 0.32], I²=60.84%, P value=0.58, Corrected P value=0.75) (Fig 8B). Four studies compared choriocapillaris flow area (3 mm) between non-ocular BD and HC subgroups utilizing the RTVue XR Avanti device [34, 35, 45, 48]. Meta-analysis (142 non-ocular BD and 153 HC eyes) demonstrated no significant difference between the two groups (Hedges g=0.10, CI = [−0.51 to 0.70], I²=84.75%, P value=0.75, Corrected P value=0.75) (Fig 8C).

   **3.5.11. FAZ** (non-ocular BD vs. HC). Ten studies compared FAZ between Non-ocular BD and HC groups using RTVue XR Avanti [35–37, 40, 45, 47–49] and Heidelberg [28, 42] devices. Pooled results (398 non-ocular BD and 381 HC eyes) demonstrated a trend toward significant difference in the non-ocular BD group compared to HCs (Hedges g=0.82, CI= [0.01 to 1.63], I²=96.12%, P value=0.05, Corrected P value=0.06) (Fig 8D).

   **3.5.12. FAZ superficial** (non-ocular BD vs. HC). Six studies made a comparison for FAZ superficial among Non-ocular BD and HC groups using RTVue XR Avanti [34, 35, 38, 45, 53] and Heidelberg [28] devices. Meta-analysis of the six studies (188

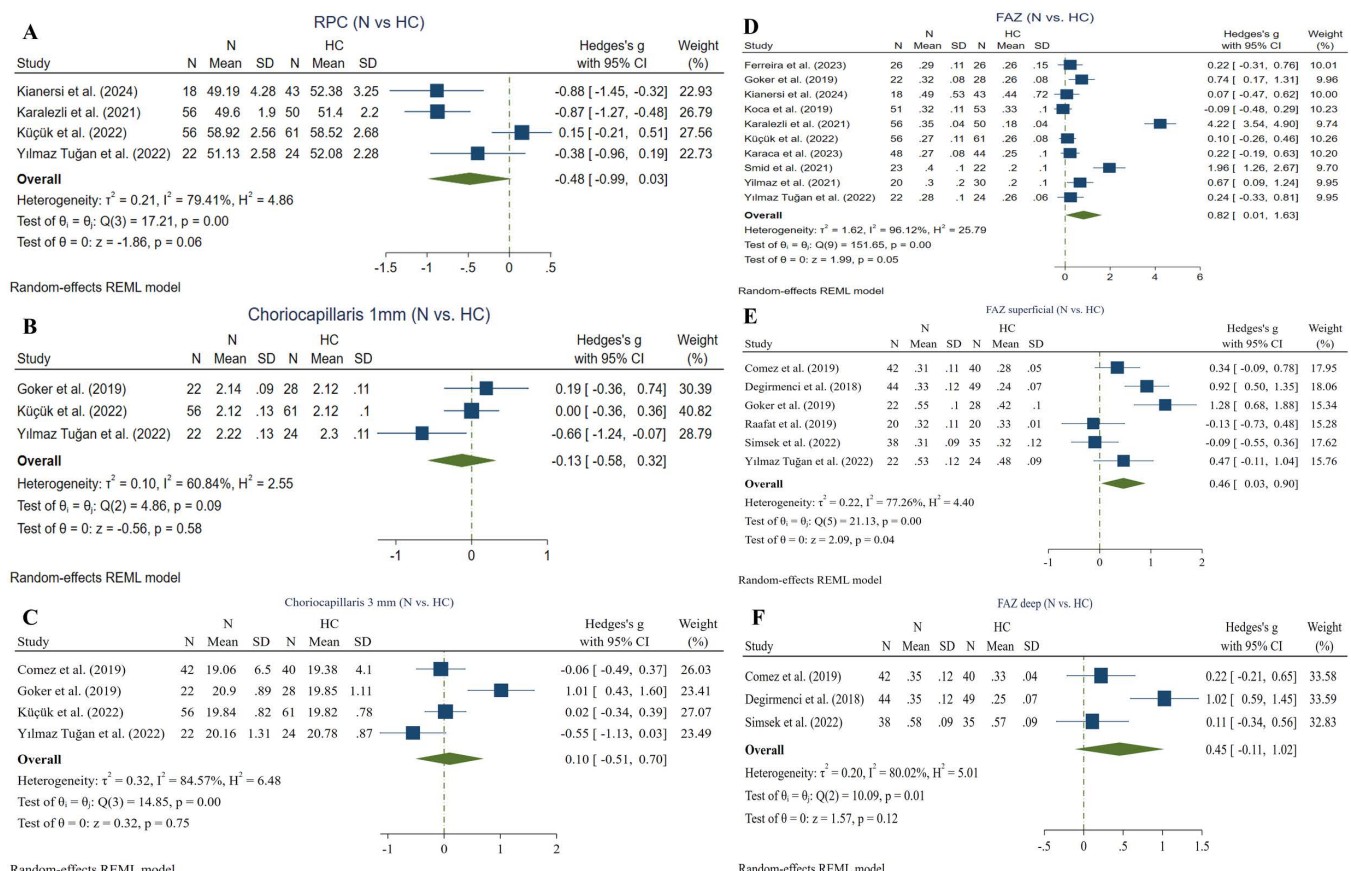

**Fig 8. Non-ocular BD eyes vs. HCs.** A: RPC VD in non-ocular BD and HC eyes, B: Choriocapillaris flow area (1 mm) in non-ocular BD and HC eyes, C: Choriocapillaris flow area (3 mm) in non-ocular BD and HC eyes, D: FAZ in non-ocular BD and HC eyes, E: Superficial FAZ in non-ocular BD and HC eyes, F: Deep FAZ in non-ocular BD and HC eyes. Abbreviations: BD: Bechet disease, CI: confidence interval, FAZ: foveal avascular zone, N: non-ocular BD, RPC: radial peripapillary capillary, SD: standard deviation, VD: vessel density.

non-ocular BD and 196 HC eyes) showed a significant increase in the non-ocular BD group compared to HCs (Hedges g = 0.46, CI= [0.03 to 0.90], $I^2$ = 77.26%, P value = 0.04, Corrected P value = 0.04) (Fig 8E).

**3.5.13. FAZ deep** (non-ocular BD vs. HC). Three studies using RTVue XR Avanti addressed the comparison of FAZ deep between Non-ocular BD and HC groups [34, 39, 53]. Meta-analysis (124 non-ocular BD and 124 HC eyes) revealed demonstrated no significant difference between the groups (Hedges g = 0.45, CI= [−0.11 to 1.02], $I^2$ = 80.02%, P value = 0.12, Corrected P value = 0.12) (Fig 8F).

## 3.6. Non-specified ocular BD vs. HC

**3.6.1. Foveal SRCP** (non-specified ocular BD vs. HC). Two studies compared SRCP foveal VD between non-specified ocular BD and HC subgroups applying the RTVue XR Avanti device [40, 49]. Meta-analysis (69 non-specified ocular BD and 73 HC eyes) showed no significant difference in the metric between the two groups (Hedges g = −0.12, CI= [−0.45 to 0.21], $I^2$ = 0%, P value = 0.46, Corrected P value = 0.46) (**Fig 9A**).

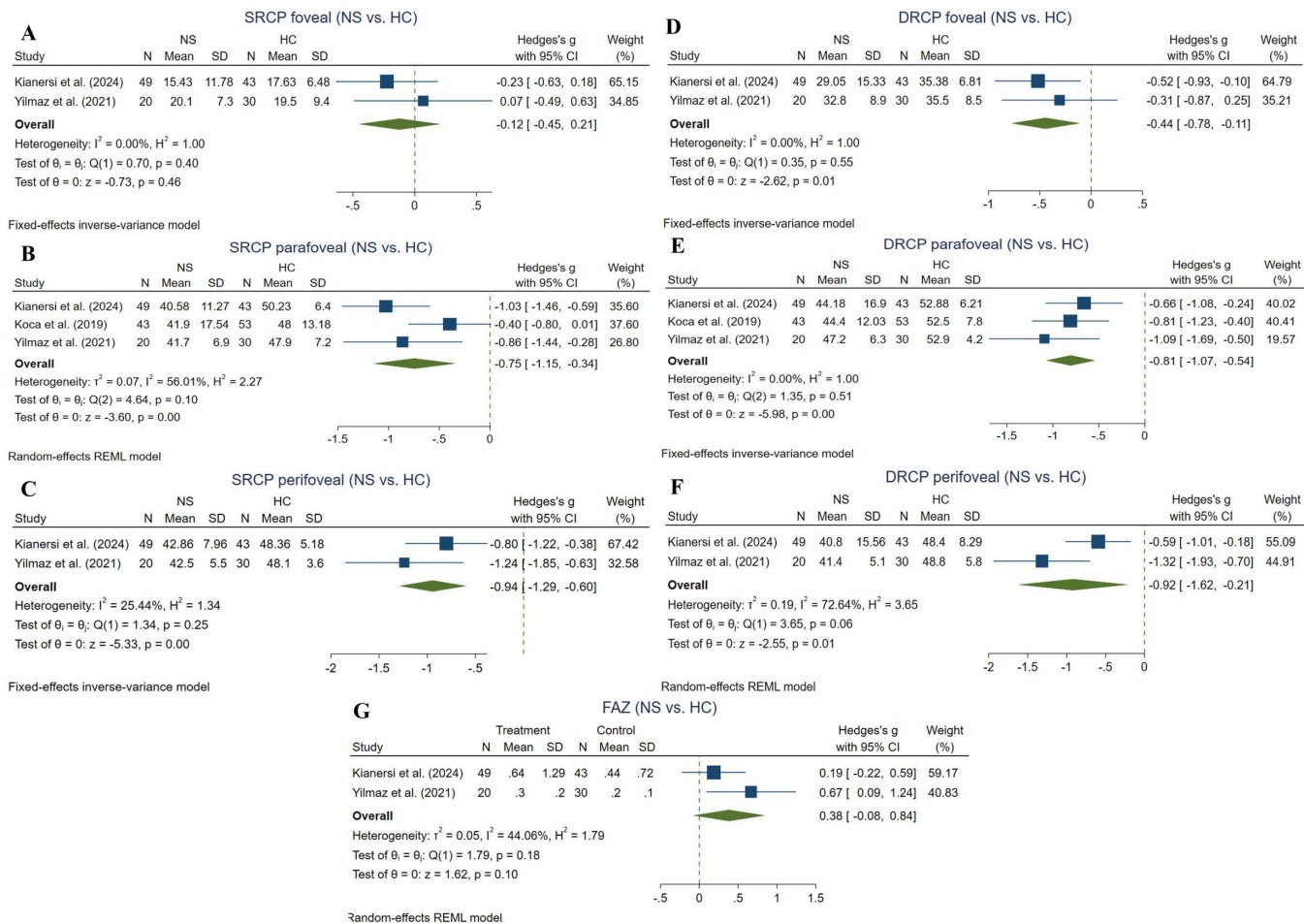

**Fig 9. Non-specified BD eyes vs. HCs.** A: Foveal SRCP VD in non-specified BD and HC eyes, B: Parafoveal SRCP VD in non-specified BD and HC eyes, C: Perifoveal SRCP VD in non-specified BD and HC eyes, D: Foveal DRCP VD in non-specified BD and HC eyes, E: Parafoveal DRCP VD in non-specified BD and HC eyes, F: Perifoveal DRCP VD in non-specified BD and HC eyes, G: FAZ in non-specified BD and HC eyes. Abbreviations: BD: Bechet disease, HC: Healthy Controls, CI: confidence interval, FAZ: foveal avascular zone, SD: standard deviation, SRCP: superficial retinal capillary plexus, DRCP: deep retinal capillary plexus, VD: vessel density.

**3.6.2. Parafoveal SRCP** (non-specified ocular BD vs. HC). Three studies compared SRCP parafoveal VD between non-specified ocular BD and HC subgroups applying the RTVue XR Avanti device [37, 40, 49]. Meta-analysis (112 non-specified ocular BD and 116 HC eyes) showed a significant reduction in the metric in the non-specified ocular BD group (Hedges g = −0.75, CI= [-.1.153 to −0.34], $I^2$= 56.01%, P value = 0.003, Corrected P value = 0.004) (**Fig 9B**).

**3.6.3. Perifoveal SRCP** (non-specified ocular BD vs. HC). Two studies compared SRCP perifoveal VD between non-specified ocular BD and HC subgroups applying the RTVue XR Avanti device [40, 49]. Meta-analysis (69 non-specified ocular BD and 73 HC eyes) showed a significant reduction in the metric in the non-specified ocular BD group (Hedges g = −0.94, CI= [−1.29 to −0.59], $I^2$= 25.44%, P value = 0.00001, Corrected P value = 0.00002) (**Fig 9C**).

**3.6.4. Foveal DRCP** (non-specified ocular BD vs. HC). Two studies compared DRCP foveal VD between non-specified ocular BD and HC subgroups applying the RTVue XR Avanti device [40, 49]. Meta-analysis (69 non-specified ocular BD and 73 HC eyes) revealed a significant difference between the two groups in favor of the HC group (Hedges g = −0.44, CI= [−0.78 to −0.11], $I^2$= 0.00%, P value = 0.008, Corrected P value = 0.012) (**Fig 9D**).

**3.6.5. Parafoveal DRCP** (non-specified ocular BD vs. HC). Three studies compared DRCP parafoveal VD between non-specified ocular BD and HC subgroups applying the RTVue XR Avanti device [37, 40, 49]. Meta-analysis (112 non-specified ocular BD and 116 HC eyes) revealed a significant difference between the two groups in favor of the HC group (Hedges g = −0.81, CI= [−1.07 to −0.54], $I^2$= 0.00%, P value = 0.00001, Corrected P value = 0.000015) (**Fig 9E**).

**3.6.6. Perifoveal DRCP** (non-specified ocular BD vs. HC). Two studies compared FAZ between non-specified ocular BD and HC subgroups applying the RTVue XR Avanti device [40, 49]. Meta-analysis (69 non-specified ocular BD and 73 HC eyes) revealed no significant difference between the two groups in favor of the HC group (Hedges g = 0.38, CI= [−0.08 to 0.84], $I^2$= 44.64%, P value = 0.1, Corrected P value = 0.1) (**Fig 9F**).

**3.6.7. FAZ** (non-specified ocular BD vs. HC). Two studies compared DRCP perifoveal VD between non-specified ocular BD and HC subgroups applying the RTVue XR Avanti device [40, 49]. Meta-analysis (69 non-specified ocular BD and 73 HC eyes) revealed a significant difference between the two groups in favor of the HC group (Hedges g = −0.92, CI= [−1.62 to −0.21], $I^2$= 72.64%, P value = 0.01, Corrected P value = 0.04) (**Fig 9G**).

### 3.7. Meta-regression

We investigated the effect of age, disease duration, BCVA, and IOP on all eligible analyses using meta-regression. However, only analyses regarding non-ocular BD and HC yielded significant associations; hence, in the following, we showcased the significant associations in this regard. SRCP parafoveal meta-analysis showed a significant positive association with disease duration (Coefficients: 0.08, P value = 0.01). DRCP whole (6*6) had a significant positive relation with the IOP of patients (Coefficients: 0.51, P value = 0.008). Significantly, foveal DRCP was negatively influenced by BCVA of controls (Coefficients: −0.82, P value = 0.03), and IOP of patients (Coefficients: 0.73, P value = 0.01. Parafoveal DRCP meta-analysis significantly negatively affected the BCVA of the control (Coefficients: −1.32, P value = 0.008). The meta-analysis of RPC VD was affected positively by non-ocular BD IOP (Coefficients: 0.48, P value <0.001). Choriocapillaris at 1 mm diameter revealed significant associations with both patient and control age (Coefficients: 0.02, 0.02, P values = 0.03). Moreover, disease duration had significant positive impacts on the analysis of both 1 mm and 3 mm choriocapillaris (Coefficients: 0.1, 0.17, respectively. P values = 0.03, 0.003, respectively). Despite insignificant results for the IOP of controls, the case's IOP significantly influenced the analysis of 3 mm choriocapillaris (Coefficients: 1.31, P values = 0.03). FAZ size showed associations with several factors. Significant negative relationships were found for FAZ analyses with IOP of subjects and controls (Coefficients: −1.9, −1.5, respectively. P values: <0.001, 0.02, respectively). However, the factor exhibited positive associations when conducting a meta-analysis for the deep portion of FAZ (Coefficients: 0.26, 0.19, respectively. P values: 0.002, 0.001, respectively). Besides, deep FAZ was influenced by other factors such as the age of subjects and controls (Coefficients: 0.08, 0.05, respectively. P values: 0.01, 0.004, respectively) and strong negative associations between deep FAZ analyses and BCVA metrics of patients and HC (Coefficients: −8.5, −7.13, respectively. P values: 0.002, 0.002, respectively).

### 3.8. Risk of bias assessment

The studies were evaluated for the risk of bias according to the NOS checklist. Articles were evaluated in three main domains, including "selection," "comparability," and "exposure". Seven studies scored 9/9 [28,35,39,42,43,48,56], six scored 8/9 [25,30,32,47,49,58], and six scored 7/9 [29,37,40,45,46,50], four scored 6/9 [27,34,38, 53], and five scored 5/9 [26, 31, 33, 36, 44]. **Table S4** in S1 File)details the quality assessment of the studies.

## 4. Discussion

Investigating the novel OCTA modality, the present systematic review and meta-analysis incorporated 28 studies, including a total of (769 BD subjects, 123 active BU eyes, 462 inactive BU eyes, 112 non-specified ocular BD, and 486 non-ocular BD eyes). The meta-analysis revealed that patients with inactive BU and non-ocular BD exhibited significantly larger FAZ sizes and lower VD in both the superficial and deep retinal capillary plexuses (SRCP and DRCP) compared to HCs, particularly in the parafoveal sector. Additionally, RPC VD was lower in inactive BU patients than in HCs. However, no significant differences in OCTA parameters were found in patients with active BU compared to HCs, nor were there significant differences in choriocapillaris flow among the groups.

One of the explanations for the latter could rely on the nature of the underlying inflamed vessel in the active phase. It seems that inflammatory processes compromise the integrity of the vessel wall, leading to plasma leakage. This, in turn, decreases blood flow to levels undetectable by OCTA [59]. Apart from plasma leakage, given the smaller capillary diameter, low-velocity blood flow in the lumen falls below the detectable range of OCTA when vasculitis occurs [28]. Another reason could be simply due to the limitations of the imaging itself. OCTA image quality would be affected by media opacity, for instance, in the presence of cells and flare in the anterior chamber or vitreous [60], which raises another hypothesis for the mentioned insignificancy.

The present study confirmed that the inactive state of BU exhibits reduced VDs, which is reasonable considering BD is a necrotizing occlusive vasculitis affecting arteries and veins. In this accordance, whether inactive BU eyes were the fellow eyes during unilateral uveitis or the resolved previously active uveitis, reduced VDs were inevitable [44]. Consistent with meta-analysis results, inactive BU eyes showed a lower RPC VD compared to the control group [29, 31], despite insignificant differences observed in the non-ocular form of the disease [45, 47, 48]. RPC VD is prone to ischemia and was shown to be impaired in BU, highlighting the disease's impact on optic nerve head circulation, which is supported by a study that showed a decrease in RPC VD contributed to poorer visual acuity [29]. In fact, the RPC network contains high metabolic demand for unmyelinated nerves, making it more susceptible to ischemia in vascular compromise conditions such as Behcet vasculitis [14].

This study postulated that BD, in the form of clinically unaffected eyes, can still induce microvascular impairment of the retina. Given that Behcet is a multisystem inflammatory disease with features of vascular obliteration [38], and due to the presence of inflammatory mediators such as C-reactive protein and erythrocyte sedimentation rate (ESR), immune complexes, and endothelial cell damage, some level of decrease in VD in retinal vascular layers has been observed in non-ocular BU [25,31,44,61,62]. Further evidence for microvascular involvement in non-ocular BD comes from the correlation between BD duration and subclinical worsening of retinal microvasculature in non-ocular BD patients using OCTA [28]. This supports the notion that BD duration itself induces microvascular alterations independent of uveitis attacks. With that in mind, an interesting extrapolation made by Smid et al. indicated that future longitudinal investigations should insist on whether non-ocular BD with detected impaired microvasculature will become more vulnerable to uveitis attacks in the future [28].

OCTA enables investigators to evaluate FAZ areas easily, in contrast to the conventional FA method in which, due to dye leakage, both eyes could not be studied simultaneously [26]. In line with the meta-analysis result, FAZ enlargement in BD, regardless of ocular involvement, is well documented in the literature [24, 25, 27, 28, 35]. Conversely, some investigations found no difference in FAZ size between BD and healthy subjects [34, 37, 38]. Several reasons could explain this

inconsistency in the literature. Firstly, between-subject variability of FAZ size is considerably high among healthy eyes, and it also varies with age, sex, and ethnicity [34, 37, 38, 63–65]. Another bias may be considering the fellow clinically healthy eyes as the non-ocular BD while the contralateral eye is involved in BU, as in the Koca et al. study [37]. Since potential circulatory inflammatory factors affect the microvasculature of the fellow eyes, it is reasonable not to allocate the fellow eyes of unilateral BU as entirely healthy ones. Altogether, this observation underscores the significance of FAZ enlargement as a marker of retinal ischemia in BD, advocating for future studies.

The present study has several limitations. 1) The observed results could be classified by various Behcet's treatment options; although treatment status was not well stated in the included studies. 2) Device-related limitations, such as possible segmentation errors and the inability to detect low flow, could also affect the outcomes. 3) Another limitation relies on the definitions of active and inactive BU. The differentiation might not be accurate as residual retinal vascular leakage may be present if the remission phase is only judged clinically. 4) The exact number of uveitis attacks and disease duration were ambiguous in studies, which can potentially affect the comparison of the studies.

## 5. Conclusions

Taken together, this meta-analysis highlights microvascular alterations in various retinal regions and different stages of BD. While our findings did not demonstrate a significant reduction in vessel density in active BD cases, this may be influenced by factors such as prior medical therapy, the varying severity of disease activity, and limitations in OCTA imaging due to poor visualization in severe cases. Future research should aim to clarify these confounding variables and explore whether other microvascular parameters may serve as more sensitive markers of disease activity.

Interestingly, the non-ocular form of BD exhibited retinal microvascular impairment in OCTA studies, suggesting that retinal screening could be beneficial for all BD patients, even in the absence of clinical ocular involvement. However, the clinical relevance of such screening remains uncertain, particularly in terms of management implications. Further longitudinal studies are required to assess the progression of vascular impairment over time and determine whether early detection through OCTA has practical consequences for patient care.

## Supporting information

**S1 File. Tables S1–S6 and Supplemental Results are compiled in this single. PDF.**
(PDF)

**S2 File. The data supporting the findings is available in the compressed supplemental file.**
(RAR)

**S3 File. PRISMA Guideline for Reporting Systematic Reviews.**
(DOCX)

**S4 File. PRISMA Guideline for Reporting Abstract of Systematic Reviews.**
(DOCX)

## Author contributions

**Conceptualization:** Mehrdad Mozafar, Mobina Amanollahi, Reza Samiee, Melika Jameie, Ali Mousavi, Zahra Ghanbari, Helia Nafar, Negar Mozafar, Fatemeh Amiri, Elias Khalili Pour, Nazanin Ebrahimiadib.

**Data curation:** Mehrdad Mozafar, Mobina Amanollahi, Reza Samiee, Melika Jameie, Ali Mousavi, Zahra Ghanbari, Helia Nafar, Negar Mozafar, Fatemeh Amiri.

**Formal analysis:** Mehrdad Mozafar, Mobina Amanollahi, Mehdi Azizmohammad Looha.

**Methodology:** Mehrdad Mozafar, Mobina Amanollahi, Mehdi Azizmohammad Looha, Elias Khalili Pour.

**Project administration:** Elias Khalili Pour.

**Supervision:** Mehrdad Mozafar, Mobina Amanollahi, Elias Khalili Pour, Nazanin Ebrahimiadib.

**Validation:** Mehrdad Mozafar, Elias Khalili Pour.

**Visualization:** Mehrdad Mozafar.

**Writing – original draft:** Mehrdad Mozafar, Mobina Amanollahi, Reza Samiee, Melika Jameie, Ali Mousavi, Zahra Ghanbari, Helia Nafar, Negar Mozafar, Fatemeh Amiri, Mehdi Azizmohammad Looha.

**Writing – review & editing:** Mehrdad Mozafar, Elias Khalili Pour, Nazanin Ebrahimiadib.

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
