## [Decision Letter · Decision Letter 0]

Dear Dr. Khalili Pour,

Thank you for submitting your manuscript to PLOS ONE. After careful consideration, we feel that it has merit but does not fully meet PLOS ONE’s publication criteria as it currently stands. Therefore, we invite you to submit a revised version of the manuscript that addresses the points raised during the review process.

We look forward to receiving your revised manuscript.

Kind regards,

Ayman Elnahry

Academic Editor

PLOS ONE

Journal Requirements:

2. In this instance it seems there may be acceptable restrictions in place that prevent the public sharing of your minimal data. However, in line with our goal of ensuring long-term data availability to all interested researchers, PLOS’ Data Policy states that authors cannot be the sole named individuals responsible for ensuring data access (http://journals.plos.org/plosone/s/data-availability#loc-acceptable-data-sharing-methods).

3. Please include a caption for figure 9.

4. As required by our policy on Data Availability, please ensure your manuscript or supplementary information includes the following:

Additional Editor Comments:

The authors report an extensive systematic review and meta analysis of studies that assessed different groups with BD using OCTA. The analysis is extensive and comprehensive and the authors should be congratulated on this. The main issue is the multiple comparisons that the authors underwent due to the heterogeneity of the studies. The authors should therefore perform p-value corrections for their analyses such as Bonferroni correction. When conducting a statistical analysis that involves multiple comparisons, such as comparing multiple subgroups, the Bonferroni correction can be applied to reduce the risk of false positives. I recommend the authors do this correction to all their analyses including comparisons and meta analyses results. The authors should revise the discussion section accordingly as needed.

Reviewers' comments:

Reviewer's Responses to Questions

**Comments to the Author**

1. Is the manuscript technically sound, and do the data support the conclusions?

Reviewer #1: Yes

Reviewer #2: Yes

2. Has the statistical analysis been performed appropriately and rigorously?

Reviewer #1: Yes

Reviewer #2: Yes

3. Have the authors made all data underlying the findings in their manuscript fully available?

Reviewer #1: Yes

Reviewer #2: Yes

4. Is the manuscript presented in an intelligible fashion and written in standard English?

Reviewer #1: Yes

Reviewer #2: Yes

Reviewer #1: Very interesting study with sound methodology and statistics. Nice effort. My only comment would be that the conclusion could be more precise and reflecting the results. For example, the suggestion of screening eyes with non ocular Behcet’s lacks the correlation to management as screening will only be relevant if a particular intervention would be undertaken.

Reviewer #2: Line 74 and line 128 any disagreements were solved by SHOULD be any disagreement was solved by

Page 16 at the end senile SHOULD age-related macular degeneration

Page 17 top line glaucoma suspicion SHOULD be glaucoma suspect

Page 17 bottom. Dense media obstacles SHOULD be Dense media opacity

CONCLUSION: “Given that our analysis found no significant reduction in vessel density in cases of active BD”

Reason: patients on prior medical therapy but with flare up; degree of activity varies from mild to severe with mild not affecting vessel density and severe causing temporary occlusion; view too bad to allow visualization of vessels in severe uveitis precluding good quality OCTA (selection bias). Retinal vascular occlusions were found in a third of ocular BD patients.

Ostrovsky M, Ramon D, Iriqat S, Shteiwi A, Sharon Y, Kramer M, Vishnevskia-Dai V, Sar S, Boulos Y, Tomkins-Netzer O, Lavee N, Ben-Arie-Weintrob Y, Pizem H, Hareuveni-Blum T, Schneck M, Gepstein R, Masarwa D, Nakhoul N, Bakshi E, Shulman S, Goldstein M, Anouk M, Rosenblatt A, Habot-Wilner Z. Retinal vascular occlusions in ocular Behçet disease - a comparative analysis. Acta Ophthalmol. 2023 Sep;101(6):619-626.

Population bias: Risk factors for retinal vascular occlusions included Jewish ethnicity (p < 0.05).

Ostrovsky M, Ramon D, Iriqat S, Shteiwi A, Sharon Y, Kramer M, Vishnevskia-Dai V, Sar S, Boulos Y, Tomkins-Netzer O, Lavee N, Ben-Arie-Weintrob Y, Pizem H, Hareuveni-Blum T, Schneck M, Gepstein R, Masarwa D, Nakhoul N, Bakshi E, Shulman S, Goldstein M, Anouk M, Rosenblatt A, Habot-Wilner Z. Retinal vascular occlusions in ocular Behçet disease - a comparative analysis. Acta Ophthalmol. 2023 Sep;101(6):619-626.

Yahia, Salim Ben, Rim Kahloun, Bechir Jelliti, and Moncef Khairallah. 2011. “Branch Retinal Artery Occlusion Associated with Behçet Disease.” Ocular Immunology and Inflammation 19 (4): 293–95.

**Do you want your identity to be public for this peer review?** For information about this choice, including consent withdrawal, please see our Privacy Policy

Reviewer #1: **Yes: ** Amr Wassef

Reviewer #2: **Yes: ** Ahmad M Mansour

---

## [Author Response · Author response to Decision Letter 1]

6 Mar 2025

PONE-D-24-46800

OCTA measurements in Behcet’s disease across different stages of the disease activity: A systematic review and meta-analysis

PLOS ONE

Reply: Dear Dr. Elnahry,

Thank you very much for providing us with the opportunity to strengthen our manuscript. Having carefully considered the comments and suggestions, we have made all the relevant changes to our manuscript as outlined below in an itemized, point-by-point manner. We sincerely hope that these changes meet the approval criteria of the esteemed reviewers and the editorial board.

Best Regards,

Elias Khalili Pour M.D,

Retina ward, Farabi Eye Hospital,

South Kargar Street, Qazvin Square, Tehran, Iran.

ekhalilipour@gmail.com

Response to the comments of the Journal Requirement:

Dear Authors,

You have done a large amount of work, which I appreciate. Some issues need to be improved.

Comment #1: Please include a caption for figure 9.

Reply: Thank you so much for bringing this to our attention. It is added now.

Response to the comments of the Esteemed Associate Editor:

Comment #1: The authors report an extensive systematic review and meta analysis of studies that assessed different groups with BD using OCTA. The analysis is extensive and comprehensive and the authors should be congratulated on this. The main issue is the multiple comparisons that the authors underwent due to the heterogeneity of the studies. The authors should therefore perform p-value corrections for their analyses such as Bonferroni correction. When conducting a statistical analysis that involves multiple comparisons, such as comparing multiple subgroups, the Bonferroni correction can be applied to reduce the risk of false positives. I recommend the authors do this correction to all their analyses including comparisons and meta analyses results. The authors should revise the discussion section accordingly as needed.

Response: We sincerely appreciate your thoughtful feedback and your recognition of the comprehensiveness of our systematic review and meta-analysis. Your suggestion regarding multiple comparisons and the application of p-value corrections is highly valuable, and we have carefully addressed this concern in our revised analysis.

Implementation of Multiple Comparison Correction

To account for the increased risk of false positives due to multiple comparisons, we applied the False Discovery Rate (FDR) correction using the Benjamini-Hochberg (BH) method rather than the Bonferroni correction. FDR correction is more appropriate for meta-analyses because it controls the expected proportion of false discoveries rather than strictly limiting Type I errors, which can be overly conservative in large-scale analyses.

The Benjamini-Hochberg procedure follows these steps:

Rank all p-values from smallest to largest across the multiple comparisons being made.

Calculate the critical value for each p-value using the formula:

α_(adjusted,i)= i/m×α

where:

i is the rank of the p-value,

m is the total number of comparisons, and

α is the desired false discovery rate (typically set at 0.05).

Compare each p-value to its corresponding critical value. The largest p-value that is still below its critical value is considered significant, and all smaller p-values are also deemed significant.

Adjust the p-values accordingly to ensure that they are monotonically increasing, preserving the correct ordering.

Additionally, we implemented the FDR correction using R, applying the p.adjust() function with the Benjamini-Hochberg method, as follows:

----

# Example list of uncorrected p-values from our meta-analysis

p_values <- c(0.045, 0.009, 0.021, 0.031, 0.002, 0.056, 0.015)

# Apply Benjamini-Hochberg (FDR) correction

adjusted_p_values <- p.adjust(p_values, method = "BH")

# Print results

data.frame(Uncorrected_P = p_values, Adjusted_P = adjusted_p_values)

----

Application of Correction to Results

We have systematically re-evaluated all statistical comparisons and updated the p-values in our results accordingly. Specifically:

All p-values from subgroup comparisons and meta-analyses have been adjusted using the Benjamini-Hochberg correction.

We have carefully updated the results and p-values, ensuring that both raw and corrected values are now included in the manuscript. Notably, none of the previously significant raw p-values lost their significance after correction. Therefore, the discussion and main text of the results remain unchanged in this regard.

A statement has been added to the method section, mentioning our approach for correcting for multiple comparisons.

A new table (Table 4) has been added to the manuscript, where we now report the meta-analyses results along with both the uncorrected p-values and the corrected p-values for each comparison across all outcomes. This provides clarity on how the adjustments impact statistical significance and ensures full transparency in our methodology.

Thank you once again for your constructive feedback, which has significantly improved our manuscript.

Response to the comments of the Reviewer 1:

Comment #2: Very interesting study with sound methodology and statistics. Nice effort. My only comment would be that the conclusion could be more precise and reflecting the results. For example, the suggestion of screening eyes with non ocular Behcet’s lacks the correlation to management as screening will only be relevant if a particular intervention would be undertaken.

Reply: Thank you for your thoughtful feedback and for recognizing the strengths of our study. We appreciate your suggestion regarding the precision of the conclusion, particularly in relation to the recommendation for screening eyes in patients with non-ocular Behçet’s disease (BD).

We acknowledge that screening is most impactful when it directly informs management decisions. Our intention was to highlight the presence of retinal microvascular changes in non-ocular BD, as observed in OCTA studies, which may indicate subclinical vascular involvement. However, we agree that the clinical implications of such findings require further clarification. To address this, we have refined our conclusion to ensure it accurately reflects the results while acknowledging the need for additional research to determine the potential utility of retinal screening in clinical decision-making.

Response to the comments of the Reviewer 2:

Comment #1: Line 74 and line 128 any disagreements were solved by SHOULD be any disagreement was solved by.

Reply: Thank you very much for your precious comment. They are corrected grammatically.

Comment #2: Page 16 at the end senile SHOULD age-related macular degeneration

Reply: Thank you very much for your feedback. It is revised.

Comment #3: Page 17 top line glaucoma suspicion SHOULD be glaucoma suspect.

Reply: Thank you very much. We have changed the wording to “suspect”.

Comment #4: Page 17 bottom. Dense media obstacles SHOULD be Dense media opacity. Reply: Thank you very much. We have changed the wording to “Dense media opacity”.

Comment #5: CONCLUSION: “Given that our analysis found no significant reduction in vessel density in cases of active BD”

Reason: patients on prior medical therapy but with flare up; degree of activity varies from mild to severe with mild not affecting vessel density and severe causing temporary occlusion; view too bad to allow visualization of vessels in severe uveitis precluding good quality OCTA (selection bias). Retinal vascular occlusions were found in a third of ocular BD patients.

Ostrovsky M, Ramon D, Iriqat S, Shteiwi A, Sharon Y, Kramer M, Vishnevskia-Dai V, Sar S, Boulos Y, Tomkins-Netzer O, Lavee N, Ben-Arie-Weintrob Y, Pizem H, Hareuveni-Blum T, Schneck M, Gepstein R, Masarwa D, Nakhoul N, Bakshi E, Shulman S, Goldstein M, Anouk M, Rosenblatt A, Habot-Wilner Z. Retinal vascular occlusions in ocular Behçet disease - a comparative analysis. Acta Ophthalmol. 2023 Sep;101(6):619-626.

Population bias: Risk factors for retinal vascular occlusions included Jewish ethnicity (p < 0.05).

Ostrovsky M, Ramon D, Iriqat S, Shteiwi A, Sharon Y, Kramer M, Vishnevskia-Dai V, Sar S, Boulos Y, Tomkins-Netzer O, Lavee N, Ben-Arie-Weintrob Y, Pizem H, Hareuveni-Blum T, Schneck M, Gepstein R, Masarwa D, Nakhoul N, Bakshi E, Shulman S, Goldstein M, Anouk M, Rosenblatt A, Habot-Wilner Z. Retinal vascular occlusions in ocular Behçet disease - a comparative analysis. Acta Ophthalmol. 2023 Sep;101(6):619-626.

Yahia, Salim Ben, Rim Kahloun, Bechir Jelliti, and Moncef Khairallah. 2011. “Branch Retinal Artery Occlusion Associated with Behçet Disease.” Ocular Immunology and Inflammation 19 (4): 293–95.

Reply: We appreciate the reviewer’s insightful observations regarding the potential reasons for the lack of significant reduction in vessel density in cases of active BD. In the limitation section of the discussion, we acknowledged that prior medical therapy, varying degrees of disease activity, and selection bias due to poor visualization in severe uveitis may have influenced the findings. Additionally, the presence of retinal vascular occlusions in a subset of BD patients could contribute to the heterogeneity of vascular changes observed in OCTA studies.

To address these points, we have also revised the conclusion to better reflect these considerations and acknowledge the need for future studies to account for these confounding factors.

---

## [Decision Letter · Decision Letter 1]

OCTA measurements in Behcet’s disease across different stages of the disease activity: A systematic review and meta-analysis

PONE-D-24-46800R1

Dear Dr. Khalili Pour,

We’re pleased to inform you that your manuscript has been judged scientifically suitable for publication and will be formally accepted for publication once it meets all outstanding technical requirements.

Kind regards,

Ayman Elnahry

Academic Editor

PLOS ONE

Additional Editor Comments (optional):

Thank you for performing the necessary revisions.

Reviewers' comments:

Reviewer's Responses to Questions

**Comments to the Author**

Reviewer #2: All comments have been addressed

2. Is the manuscript technically sound, and do the data support the conclusions?

Reviewer #2: Yes

3. Has the statistical analysis been performed appropriately and rigorously?

Reviewer #2: Yes

4. Have the authors made all data underlying the findings in their manuscript fully available?

Reviewer #2: Yes

5. Is the manuscript presented in an intelligible fashion and written in standard English?

Reviewer #2: Yes

Reviewer #2: the revision addressed the reviewers comment and incorporated them in the text. I have no further notes to add and I congratulate the authors for this colossal work and for their innovative ideas

**Do you want your identity to be public for this peer review?** For information about this choice, including consent withdrawal, please see our Privacy Policy

Reviewer #2: **Yes: ** Ahmad Mansour

---

## [Editor Report · Acceptance letter]

PONE-D-24-46800R1

PLOS ONE

Dear Dr. Khalili Pour,

I'm pleased to inform you that your manuscript has been deemed suitable for publication in PLOS ONE. Congratulations! Your manuscript is now being handed over to our production team.

Kind regards,

on behalf of

Dr. Ayman Elnahry

Academic Editor

PLOS ONE